# CED-3 caspase acts with miRNAs to regulate non-apoptotic gene expression dynamics for robust development in *C. elegans*

**Benjamin P Weaver[1]\*[†], Rebecca Zabinsky[1][†], Yi M Weaver[2], Eui Seung Lee[1], Ding Xue[1], Min Han[2]**

[1]Department of Molecular, Cellular and Developmental Biology, University of Colorado Boulder, Boulder, United States; [2]Department of Molecular, Cellular and Developmental Biology, Howard Hughes Medical Institute, University of Colorado Boulder, Boulder, United States

**Abstract** Genetic redundancy and pleiotropism have limited the discovery of functions associated with miRNAs and other regulatory mechanisms. To overcome this, we performed an enhancer screen for developmental defects caused by compromising both global miRISC function and individual genes in *Caenorhabditis elegans*. Among 126 interactors with miRNAs, we surprisingly found the CED-3 caspase that has only been well studied for its role in promoting apoptosis, mostly through protein activation. We provide evidence for a non-apoptotic function of CED-3 caspase that regulates multiple developmental events through proteolytic inactivation. Specifically, LIN-14, LIN-28, and DISL-2 proteins are known miRNA targets, key regulators of developmental timing, and/or stem cell pluripotency factors involved in miRNA processing. We show CED-3 cleaves these proteins in vitro. We also show CED-3 down-regulates LIN-28 in vivo, possibly rendering it more susceptible to proteasomal degradation. This mechanism may critically contribute to the robustness of gene expression dynamics governing proper developmental control.

**\*For correspondence:**
benjaminpweaver@gmail.com

[†]These authors contributed equally to this work

**Competing interests:** The authors declare that no competing interests exist.

## Introduction

The robustness of animal development is ensured by multiple regulatory mechanisms with overlapping roles acting on specific cellular processes, often manifested as genetic redundancy (*Fay et al., 2002*; *Kitano, 2004*; *Felix and Wagner, 2008*; *Hammell et al., 2009*). miRNAs mostly exert repression of gene expression by blocking target mRNA translation and/or through mRNA decay as part of the miRNA-induced-silencing complex (miRISC), which includes GW182 and argonaute proteins (*Ding and Han, 2007*; *Fabian and Sonenberg, 2012*). miRNA-mediated gene silencing is a critical regulatory mechanism that ensures dynamic changes in gene expression during animal development or other physiological processes (*Ambros, 2004*; *Bartel and Chen, 2004*). However, specific physiological roles of individual miRNAs are often executed through the combinatory effects of multi-miRNA, multi-target mRNA networks (*Brenner et al., 2010*; *Karp et al., 2011*; *Kudlow et al., 2012*; *Miska et al., 2007*; *Parry et al., 2007*; *Than et al., 2013*; *Alvarez-Saavedra and Horvitz, 2010*). Moreover, these miRNA–mRNA interaction networks may act in concert, and often semi-redundantly, with other regulatory mechanisms to limit the expression of many genes involved in animal development or other physiological functions (*Figure 1A*). Therefore, tackling genetic redundancy would be critical to uncover many specific functions associated with miRNAs and other gene expression regulatory mechanisms.

**eLife digest** For an organism to develop from a single cell into a collection of many different, specialized cells, different genes must be switched on or off at particular times. However, some of these genes involved in development are 'redundant' and carry out the same or similar tasks. This acts like a backup system, so if one of the genes is unable to complete a task, the others can compensate and the organism will still develop correctly.

To produce a protein from a gene, the DNA sequence that makes up the gene is used as a template to create another molecule called messenger RNA. Genes can also be 'silenced'— prevented from making proteins—by small molecules called microRNAs, which bind to messenger RNA molecules and mark them for destruction. MicroRNA molecules therefore play an important role in controlling development. However, as many microRNA molecules often work together, and as many genes are redundant, it can be difficult to discover the effects of specific microRNAs. It is also difficult to discover whether any other mechanisms work alongside the microRNAs to control development.

Weaver, Zabinsky et al. used mutant forms of the nematode worm *Caenorhabditis elegans*, in which microRNA gene regulation did not work correctly, to investigate the mechanisms that work alongside microRNAs to control development. Genes in these worms were silenced; those silenced genes that caused additional developmental defects were considered likely to work 'redundantly' in the same role as a microRNA molecule. This revealed over one hundred genes that were previously unknown to work with microRNA molecules.

Weaver, Zabinsky et al. focused on one of these genes, called *ced-3*. The CED-3 protein produced from this gene is known to execute programmed cell death, a carefully controlled process also known as apoptosis, but was not known to have other developmental functions. However, the worms with mutant forms of the *ced-3* gene already have problems performing apoptosis but are otherwise relatively normal, so Weaver, Zabinsky et al. reasoned that the CED-3 protein must also have another role in development.

Further investigation revealed that *ced-3* mutations most severely disrupt development when they are combined with mutations in one particular family of microRNAs. These microRNAs are particularly important for controlling both when cells specialize into a particular type of cell, and the timing of when certain stages of development happen. Experiments using purified proteins showed that CED-3 breaks down three proteins that are produced from genes controlled by this family of microRNA molecules, and one of these proteins was also broken down by CED-3 in experiments with mutant worms. Weaver, Zabinsky et al. therefore propose that CED-3 is part of a semi-redundant system that ensures the proteins are produced at the right level and at the right time even if the microRNAs insufficiently regulate them. This finding demonstrated both a specific role and specific targets for the CED-3 protein during development, entirely distinct from its role in apoptosis.

Although Weaver, Zabinsky et al. have identified a large number of genes that work alongside microRNAs to control development, these are only the genes that cause obvious developmental defects in healthy worms. Further experiments using similar techniques performed on worms under stress may reveal yet more such genes.

We have carried out a genome-wide enhancer screen for genes that when knocked down would generate a strong developmental defect when general miRISC function is compromised. Among a large number of interactors identified from the screen is the *ced-3* gene that encodes a caspase, well-characterized as a key component of the apoptotic pathway (*Conradt and Xue, 2005*). While *ced-3* is absolutely required for the apoptotic process, null mutations of the gene are not associated with obvious developmental defects (*Hengartner, 1997*). However, two recent studies have reported different non-apoptotic roles of the *ced-3* pathway, namely in stress-related neuronal function (*Pinan-Lucarre et al., 2012*) and aging (*Yee et al., 2014*). Because no specific downstream targets of CED-3 were found in these studies, the mechanistic detail of such non-apoptotic functions of CED-3 remains unclear. Moreover, whether the CED-3 system is widely utilized to regulate animal development and other functions is a question of high significance.

# Results

## A genome-wide enhancer screen to identify factors that act with miRISCs to ensure robust development

To uncover specific physiological functions of miRNAs and other regulatory mechanisms acting with miRNAs during development, we performed a genetic enhancer screen for developmental defects that manifested only when miRISC function and another regulatory mechanism were both compromised (*Figure 1A*). We chose to use loss-of-function *(lf)* mutations of the *ain-1* and *ain-2* genes (GW182 orthologs) that each alone significantly compromises but does not eliminate global miRISC function (*Ding et al., 2005*; *Zhang et al., 2007*, *2009*). While the *ain-1(lf)* mutant has a mild heterochronic phenotype and the *ain-2(lf)* mutant is superficially wild-type, loss of both genes results in severe pleiotropic defects including alteration in temporal cell fate patterning. Therefore, an enhancer screen using the *ain-1(lf)* or *ain-2(lf)* mutant can potentially detect functions associated with most miRNAs.

Using the entire *Caenorhabditis elegans* ORFeome RNAi feeding library (*Rual et al., 2004*), we performed a double-blind screen that identified 126 genetic interactors (*Figure 1A–D*, *Figure 1—figure supplement 1* and *Supplementary files 1,2*), of which only eight have been reported to interact with miRNA regulatory pathways (*Parry et al., 2007*). Many interactions were confirmed by testing candidate mutants for phenotypes when treated with *ain-1* and *ain-2* RNAi (*Supplementary file 3*). Nearly two-thirds of the 126 genetic interactors were found to interact with both *ain-1* and *ain-2* genes (*Figure 1B*). Gene ontology analysis revealed that these genes belong to a broad range of functional groups (*Figure 1C*). Over-representation of genes associated with protein stability is consistent with

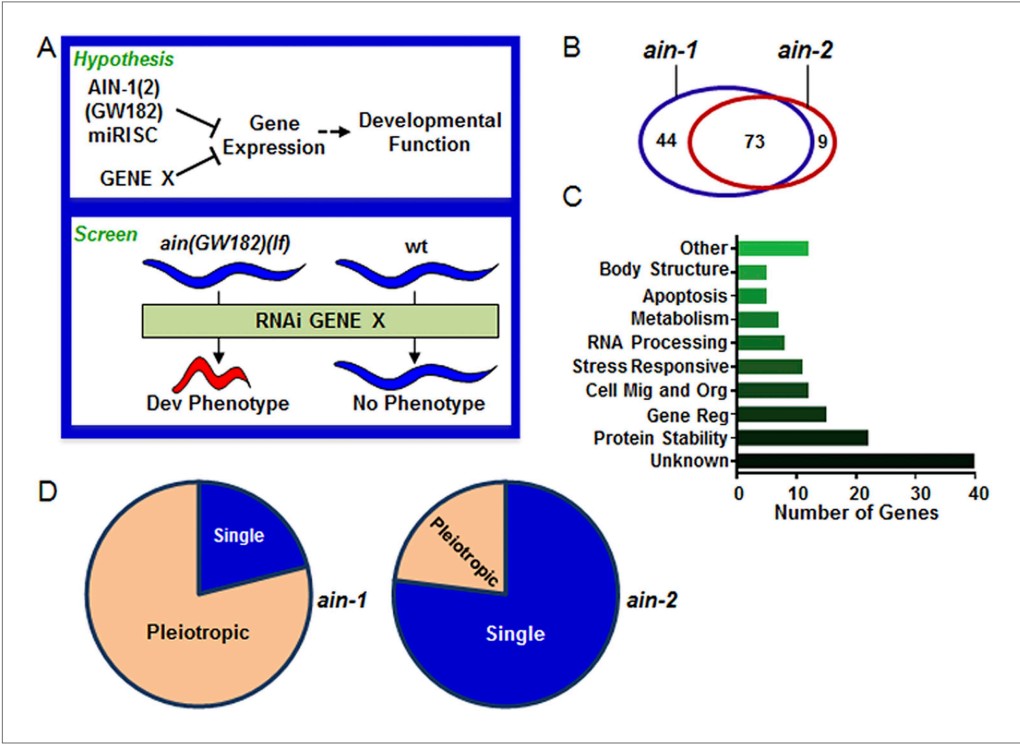

**Figure 1**. Genome-wide RNAi screen for genes that cooperate with miRISCs to regulate development. (**A**) Rationale of enhancer screen strategy (detailed in *Figure 1—figure supplement 1A*). (**B**) The number of genes identified as interactors of either/both *ain-1(lf)* and/or *ain-2(lf)*. (**C**) Distribution of the 126 interactors into functional categories (interactors listed in *Supplementary file 2*). (**D**) The proportion of genes exhibiting singular vs pleiotropic RNAi phenotypes with *ain-1(lf)* or *ain-2(lf)* (detailed phenotypic frequencies shown in *Figure 1—figure supplement 1B*).
The following figure supplement is available for figure 1:

**Figure supplement 1**. RNAi screen strategy and frequencies of phenotypes.

the hypothesis that miRNAs act in concert with other repressive mechanisms to limit gene expression (*Figure 1A,C*). We found that *ain-1(lf)* displayed more pronounced pleiotropism with its interactors than *ain-2(lf)* (*Figure 1D*) and that the two GW182 homologs have distinct frequencies of phenotypes with their interactors (*Figure 1—figure supplement 1B*), arguing against general sickness being the cause for the enhancement (further elaborated in *Figure 2—figure supplement 1*). The pleiotropic nature of *ain-1* interactions is consistent with the diverse physiological functions associated with AIN-1 or possibly its expression patterns or levels.

## Cooperation between the CED-3 pathway and miRISC on multiple aspects of development

We were most surprised to identify the *C. elegans* cell-killing caspase, *ced-3*, as an interactor of the miRISC GW182 homolog, *ain-1*. Using multiple alleles of each gene, we found that *ced-3(lf);ain-1(lf)* double mutants have pleiotropic developmental phenotypes including delays in larval growth rate, smaller brood size, abnormal adult body morphology, egg-laying defect (accumulation of eggs inside the animal), sluggish movement, embryonic lethality, and laid oocytes (failure to fertilize) (*Figure 2A–D* and *Figure 2—figure supplement 2A,B*). The penetrance of abnormal phenotypes increased as the adults continued to age (*Figure 2—figure supplement 2C*) and was therefore best quantified in a synchronized population. Combining mutations of miRISC components such as *ain-1(GW182)(lf)* or *alg-1(argonaute)(lf)* with the cell death pathway factors *ced-3(caspase)(lf)* or its upstream activator, *ced-4(apaf-like)(lf)*, results in abnormal adults (*Figure 2E*) but *ced-3(lf);ain-2(lf)* animals did not show a significant defect (*Figure 2—figure supplement 2D*). To test the involvement of other core cell death pathway factors, we also examined the interaction of *ain-1* with *egl-1* that has been shown to act upstream of the CED-3 caspase to promote apoptosis (*Figure 2—figure supplement 3A*) and *egl-1(lf)* is known to cause a strong cell death defect (*Conradt and Xue, 2005*). We found that, like *ced-3(lf)* and *ced-4(lf)*, *egl-1(RNAi)* also significantly enhanced the developmental defects of *ain-1(lf)* (*Figure 2—figure supplement 3B*).

To better characterize these defects, we tested the interaction in specific tissues. Expressing either *ain-1* or *ain-2* in the intestine or hypodermis alone partially rescued the defects of the *ced-3(lf);ain-1(lf)* double mutant (*Figure 2F*). These findings suggest that these two tissues are the major sites for miRNA functions in this interaction and likely also CED-3 function given that *ced-3* acts cell autonomously (*Yuan and Horvitz, 1990*). Expressing *ced-3* with strong tissue-specific promoters has been shown to kill those tissues, even in cells that do not normally die, due to the resulting high level of CED-3 accumulation (*Shaham and Horvitz, 1996*; *Hengartner, 1997*) thus preventing the reciprocal rescue experiments.

## Non-apoptotic functions of *ced-3* caspase in development

The *ced-3* caspase has been well-characterized for its role in apoptosis but not demonstrated to have a broad, non-apoptotic function in development (*Yuan et al., 1993*; *Xue et al., 1996*; *Conradt and Xue, 2005*; *Peden et al., 2008*). The fact that strong *ced-3(lf)* alleles cause robust defects in programmed cell death but not the developmental defects described above suggests that the functions of *ced-3* with miRISCs uncovered in our screen are non-apoptotic. To further address this question, we first used an assay previously shown to effectively identify apoptotic functions of genes, such as *mcd-1* encoding a zinc-finger containing protein, for which mutations caused subtle apoptotic defects alone, but significantly enhanced the cell death defect of a *ced-3* reduction-of-function allele (*ced-3(rf)*) (*Reddien et al., 2007*) (*Figure 3A*). We found that, in contrast to the positive control, *mcd-1(lf)*, the *ain-1(lf)* mutation did not enhance the apoptotic defect of *ced-3(rf)* animals as assayed by observing the perdurance of *lin-11::GFP* positive undead P9-11.aap cells (*Figure 3A–B*). Because *nuc-1* encodes an effector nuclease important for the proper execution of apoptosis (*Wu et al., 2000*), we then tested if the *ain-1(lf)* mutation was able to enhance any subtle *nuc-1(lf)* phenotype and found no significant defect beyond the phenotypes of the single mutants (*Figure 3C*). Finally, *ain-1(RNAi)* did not affect the number of apoptotic cell corpses accumulating in the heads of *ced-1(lf)* first stage larvae (*Figure 3D*), which are defective in cell corpse engulfment allowing for visualization of dead cell corpses. Therefore, the *ain-1* and *ced-3* interaction described above is non-apoptotic.

## Function of *ced-3* caspase in temporal cell fate patterning

Further analysis indicated that the *ced-3(lf)* and *ced-4(lf)* single mutants have mild reduction in their rates of post-embryonic growth similar to the *ain-1(lf)* and *alg-1(lf)* mutants (*Figure 4A–C* and also

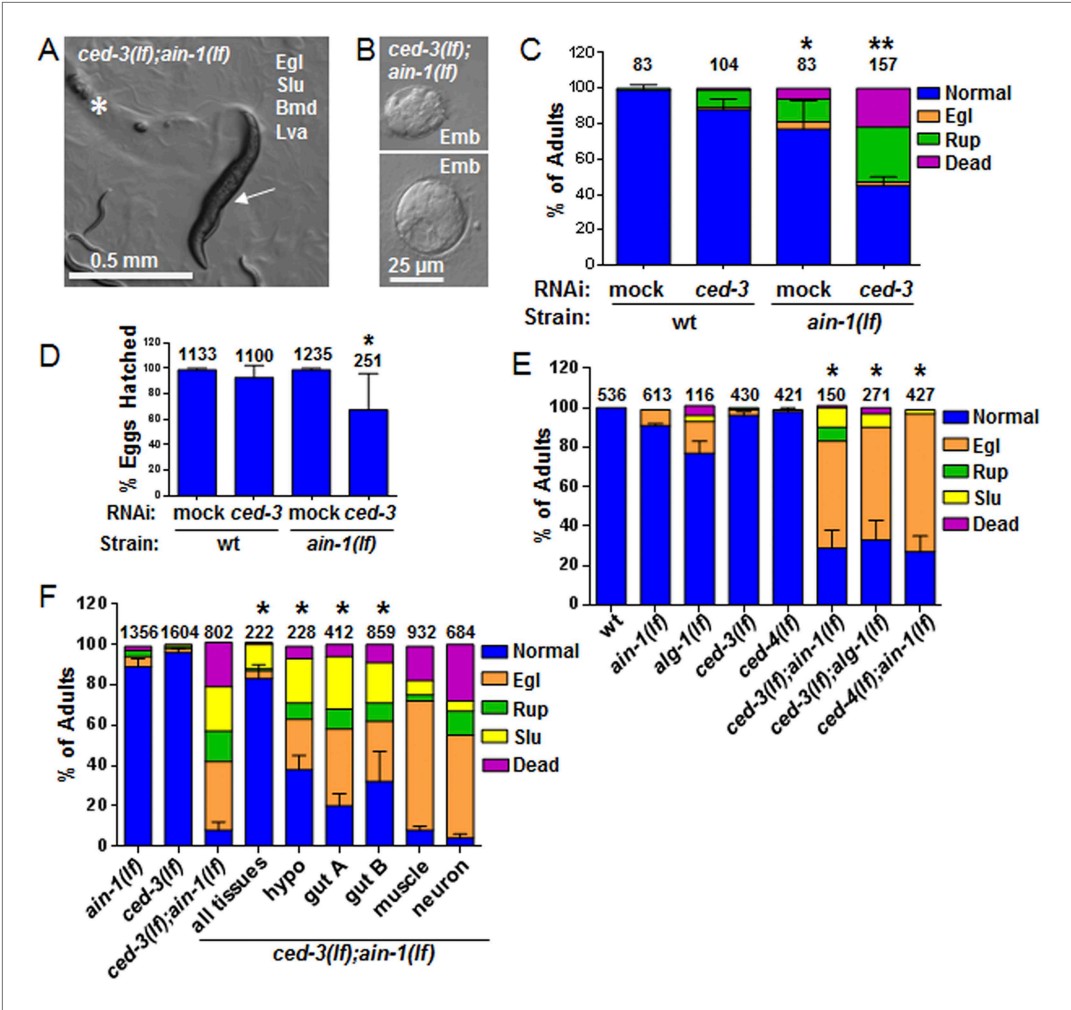

**Figure 2**. *C. elegans* strains compromised in both miRISC and *ced-3* functions have significant pleiotropic developmental phenotypes. (**A** and **B**) Microscopic images showing the pleiotropic phenotypes of the *ced-3(lf); ain-1(lf)* double mutant, including egg-laying defect (Egl), sluggish movement (Slu), body morphology defects (Bmd), larval arrest (Lva), and embryonic lethality (Emb). Asterisk in (**A**) indicates an Egl animal that was devoured by internally hatched progeny, and the arrow indicates an adult animal with multiple defects (Egl, Slu and Bmd). *Figure 2—figure supplement 1* shows the phenotype of another interactor, *ceh-18*, which is very different from *ced-3*, supporting distinct physiological relevance of the identified interactors. (**C**) *ced-3(RNAi)* significantly enhanced the frequency of *ain-1(lf)* phenotypes. Mean values ± SD for percent normal (p < 0.001, *compared to wt with mock RNAi, **compared to all others, Chi-square test comparing the distributions of phenotypes). Number of worms tested indicated above each bar (same for all figures). (**D**) Mean values ± SD of embryonic lethality (p < 0.05 **compared to all, Mann–Whitney test). (**E**) Enhancement of miRISC phenotypes by *ced-3(lf)* and *ced-4(lf)*. Mean values ± SD for percent normal (p < 0.0001, *compared to each of the relevant single mutants, Chi-square test comparing the distributions of phenotypes). Other *ain-1* and *ced-3* alleles (*Figure 2—figure supplement 2*) and the *ain-1* interaction with *egl-1* (*Figure 2—figure supplement 3*) were also tested. (**F**) Rescue effects of expressing *ain-1* or *ain-2* in specific tissues (driven by tissue-specific promoters for the four principal tissues of *C. elegans* including the hypodermis, gut, muscle, and nerve; see 'Materials and methods') in the *ced-3(lf);ain-1(lf)* double mutants. 'All tissues' indicates a genomic *ain-1* transgene. Mean values ± SD for percent normal [p < 0.0001, Fisher's Exact test comparing the distribution of normal and abnormal animals for each rescue to *ced-3(lf);ain-1(lf)* without rescue (see 'Materials and methods' for statistical rationale)].

The following source data and figure supplements are available for figure 2:

**Source data 1**. Source data quantifying genetic interactions between the miRISC and cell death pathways.

*Figure 2. Continued on next page*

*Figure 2. Continued*

**Figure supplement 1**. *ain-1(lf);ceh-18(lf)* double mutants have reduced oocytes.

**Figure supplement 2**. Additional phenotypes of *ced-3(lf);ain-1(lf)* and test of other alleles.

**Figure supplement 3**. The core apoptotic regulatory pathway acts in parallel to miRISC for normal development.

---

*Figure 4—figure supplement 1* for more *ced-3(lf)* data). Additionally, the *ced-3(lf);ain-1(lf)* and *ced-3(lf);alg-1(lf)* double mutants, but not *ced-3(lf);ain-2(lf)*, have significantly slower growth rates beyond either single mutant (*Figure 4A–C* and *Figure 4—figure supplement 1*), suggesting cooperativity in regulating the related developmental programs.

To interrogate the genetic interaction further, we screened all of the available *C. elegans* miRNA deletion strains in the blind (*Figure 5A*, strains listed in *Supplementary file 4*) for synthetic interactions with *ced-3* by depleting *ced-3* in each miRNA mutant background by RNA interference. After finding pronounced RNAi effects associated with several miRNA deletions, we then generated double or triple mutants containing *ced-3(lf)* and the miRNA mutations, and observed phenotypes similar to those seen in *ced-3(lf);ain-1(lf)* (*Figure 5B*, and refer to *Figure 2A–E*) Specifically, mutations in the *let-7*-family members, *mir-48* and *mir-84*, had the strongest effect with a fully penetrant egg-laying defect observed in the *ced-3(lf);mir-48(lf);mir-84(lf)* triple mutant (*Figure 5B*). Interestingly, the *ced-3(lf);mir-1(lf);mir-84(lf)* triple mutant displayed some developmental defects not seen in the *mir-1(lf);mir-84(lf)*, *ced-3(lf);mir-1(lf)*, or the *ced-3(lf);mir-84(lf)* double mutants (*Figure 5B*). Since *ced-3(lf)* had the strongest developmental defects with the *let-7*-family members, and since both *lin-14* and *lin-28* mRNAs are well-known targets of the *let-7*-family of miRNAs, we thus tested the possibility that *ced-3(lf)* may enhance specific temporal cell fate patterning defects of these miRNA mutants by examining their adult alae. Normal adult-specific alae are generated by seam cells and defects in adult alae formation are commonly used as a sensitive assay for defects in temporal cell fate patterning (*Ambros and Horvitz, 1984*). We found that *ced-3(lf)* significantly enhanced adult alae defects (*Figure 5C,D*). This effect was observed for both the *miR-48(lf),miR-84(lf);ced-3(lf)* triple mutant and the *ced-3(lf);ain-1(lf)* double mutant, but not the *ced-3(lf);mir-1(lf);mir-84(lf)* triple mutant (*Figure 5D*). These findings suggested the hypothesis that the expression of some developmental timing regulators is co-regulated by miRISCs and *ced-3*.

## Negative regulation of the pluripotency factors *lin-14*, *lin-28*, and *disl-2* by *ced-3*

To better analyze the mechanism underlying this non-apoptotic temporal cell fate patterning function of *ced-3*, we tested its effect on seam cell development. The division and differentiation pattern of the stem cell-like seam cells are regulated by a well-described genetic pathway that includes several miRNAs and the LIN-28 pluripotency factor that blocks the maturation of pre-*let-7* miRNA (*Viswanathan and Daley, 2010*). During each larval stage, lateral seam cells (V1–V4 and V6) divide in an asymmetric, stem-cell like manner with additional stems cells only produced in the L2 stage by an additional symmetric division pattern that duplicates V1–V4 and V6 seam cell numbers (*Sulston and Horvitz, 1977*; *Ambros and Horvitz, 1984*). Wild-type animals consistently have 16 seam cells on both the left and right sides by adulthood (*Joshi et al., 2010*). The dynamic changes in the expression levels of several conserved pluripotency factors are critical for proper temporal cell fate patterning. LIN-14 is highly expressed during L1 to promote L1-specific developmental programs, whereas LIN-28 is highly expressed from late embryonic to L2 stages and acts to promote the L2-specific programs including the only normal symmetric division of V1–V4 and V6 seam cells (*Ambros and Horvitz, 1984*; *Ambros, 1989*; *Ruvkun and Giusto, 1989*; *Moss et al., 1997*; *Rougvie and Moss, 2013*) (Diagrammed in *Figure 6—figure supplement 2A*). Expression of LIN-14 and LIN-28 rapidly diminishes after L1 and L2, respectively, which is necessary for animals to progress to the next stage (*Figure 6—figure supplement 2A*). Loss-of-function (*lf*) mutations in *lin-14* and *lin-28* result in animals skipping the L1- and L2-specific programs, respectively (precocious phenotype) (*Figure 6—figure supplement 2A*). In contrast, hyperactive (gain-of-function, *gf*) mutations leading to prolonged expression of each gene cause the animals to reiterate the corresponding stage (retarded phenotype) (*Figure 6—figure supplement 2A*).

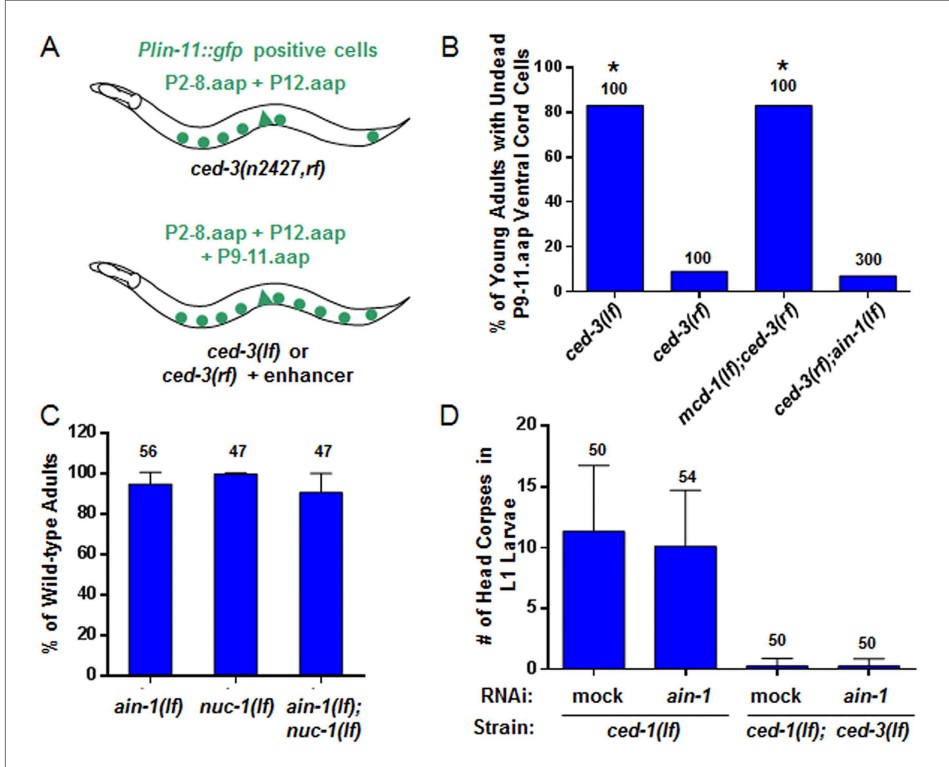

**Figure 3**. *ain-1(lf)* does not alter cell-death phenotypes. (**A**) Cartoon illustrating a previously established enhancer assay using a reduction-of-function (*rf*) *ced-3* allele (**Reddien et al., 2007**). (**B**) *ain-1(lf)* does not enhance the cell death defect of a *ced-3(rf)* mutation (p < 0.0001, *compared to *ced-3(rf)*, Mann–Whitney test). (**C**) No enhanced interaction between *ain-1(lf)* and *nuc-1(lf)*. Mean values ± SD (no significant difference, Fisher's Exact test comparing the distributions of normal and abnormal animals of the *ain-1(lf);nuc-1(lf)* double mutant to the single mutants). (**D**) *ain-1(RNAi)* does not alter apoptotic events as indicated by L1 head corpses that fail to occur in *ced-3(lf)* mutants. The *ced-1(lf)* mutation was used to enhance visualization of head corpses (**Ledwich et al., 2000**). Mean values ± SD (no significant difference, Mann–Whitney test).

The following source data is available for figure 3:

**Source data 1**. Source data quantifying apoptotic assays.

Because of the additional symmetric cell division of V1–V4 and V6 seam cells in L2, skipping or reiterating the L2 stage in *lin-28(lf)* or *lin-28(gf)* mutations lead to a decrease or increase of total seam cell number, respectively (**Ambros and Horvitz, 1984**; **Moss et al., 1997**) and diagrammed in ***Figure 6—figure supplement 2A***. Mammalian DIS3L2 was recently annotated as the ribonuclease that degrades the uridylated pre-*let-7* miRNA following binding by LIN-28 and 3′-oligo-uridylation by a polyU polymerase (**Chang et al., 2013**). We identified the likely *C. elegans* ortholog of Dis3l2 and named it *disl-2* (***Figure 6—figure supplement 1***). The effects for *disl-2* on seam cell development have not been determined.

As previously published (**Ding et al., 2005**; **Zhang et al., 2007**), we also found that the *ain-1(lf)* mutant alone has a mild increase in the number of seam cells by late larval development (***Figure 6A,B*** and ***Figure 6—figure supplement 2***) consistent with the well-established role of miRNAs in regulation of temporal cell fate patterning; whereas the *ced-3(lf)* mutant alone rarely shows altered seam cell numbers (***Figure 6A,B*** and ***Figure 6—figure supplement 2***). Strikingly, the *ced-3(lf);ain-1(lf)* double mutants have both a markedly increased number of seam cells and an increased range of seam cell number by late larval development (***Figure 6A,B***) with a mean value (±SD) of 25.9 (±5.5) per side. Notably, the *ced-3(lf);ain-1(lf)* double mutants hatch with the correct number of seam cells but they continue to increase inappropriately throughout later larval development (***Figure 6—figure supplement 2A,B***). The production of supernumerary seam cells indicates a previously unknown

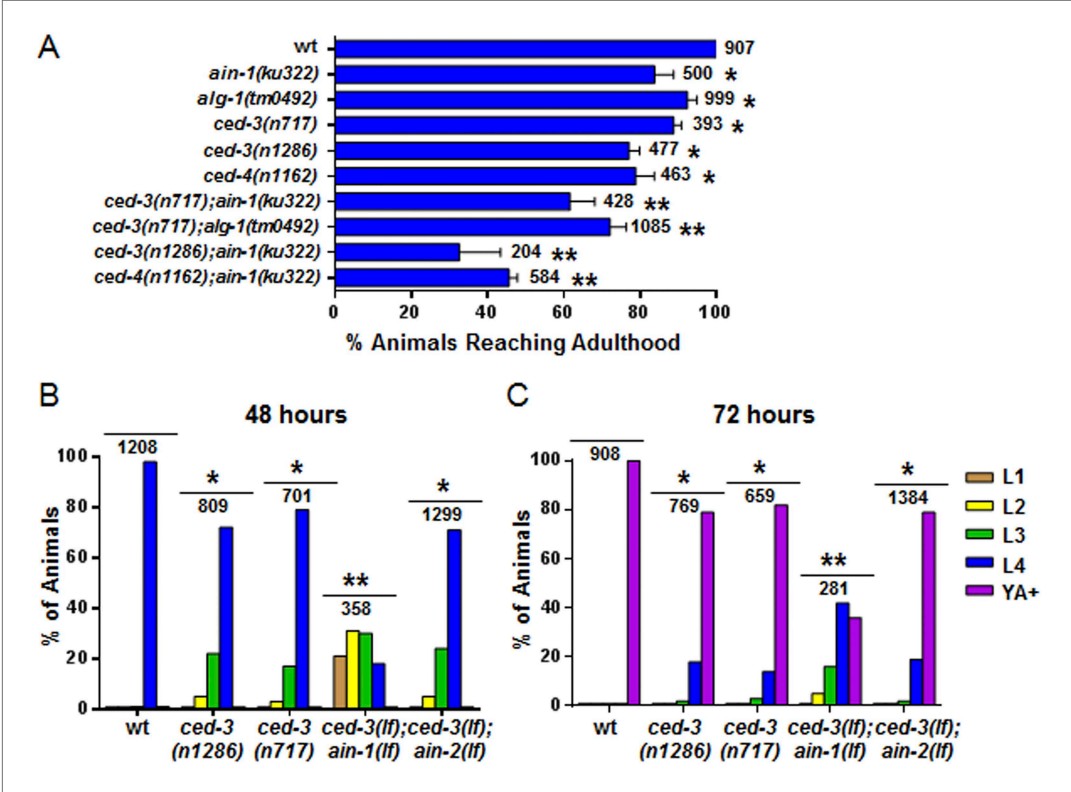

**Figure 4**. Loss of *ced-3* function slows the rate of post-embryonic development. (**A**) Percent of animals reaching adulthood at 96 hr after hatching is shown. Mean ± SD (p < 0.0001, *compared to wt, **compared to the relevant single mutants, Fisher's Exact test comparing the distributions of adult to larval-stage animals at this time). (**B** and **C**) Distribution of stages at 48 hr and 72 hr with food (p < 0.0001, *compared to wt, **compared to the relevant single mutants, Chi-square test comparing the distributions of all stages). Also see *Figure 4—figure supplement 1*.

The following source data and figure supplement are available for figure 4:

**Source data 1**. Source data quantifying post-embryonic growth rates.

**Figure supplement 1**. *ced-3(lf)* mutants displayed a mild but significant reduction in the rate of post-embryonic development.

role for *ced-3* in cooperating with miRISC-regulated seam cell differentiation and temporal cell fate patterning (*Figure 6A,B* and *Figure 6—figure supplement 2C*).

We found that the increased number of seam cells in the *ced-3(lf);ain-1(lf)* double mutants was partially suppressed by down-regulating *lin-14, lin-28,* or *disl-2(Dis3l2)* through RNAi treatment beginning at L2 (*Figure 6C*), suggesting that an abnormally high level of any of the three proteins could be a significant contributor to the phenotype. A *lin-14(lf)* or *lin-28(lf)* mutation would not be effective for such a suppression test because of the strong defects associated with them at the early larval stage (*Rougvie and Moss, 2013*). LIN-66 was previously shown to act in parallel to miRNAs to repress LIN-28 expression (*Morita and Han, 2006*). Consistent with a *ced-3* function in *lin-28*-mediated temporal cell fate patterning regulation, we also observed that *ced-3(lf)* enhanced the heterochronic defect of *lin-66* reduction (*Figure 6—figure supplement 3*). We further found that down-regulation of *lin-14, lin-28,* or *disl-2(Dis3l2)* by RNAi beginning at L2 could significantly suppress the defects in the *ced-3(lf);ain-1(lf)* double mutants (*Figure 6D*). These findings suggest that *ced-3* cooperates with miRNAs to regulate the *lin-14-lin-28-disl-2(Dis3l2)* axis during development.

## Cleavage of LIN-14, LIN-28, and DISL-2 in vitro by CED-3

The above genetic data suggest that *ced-3* normally represses *lin-28, disl-2,* and/or *lin-14* in development. As a caspase, we thought that CED-3 may directly repress the expression of these genes through

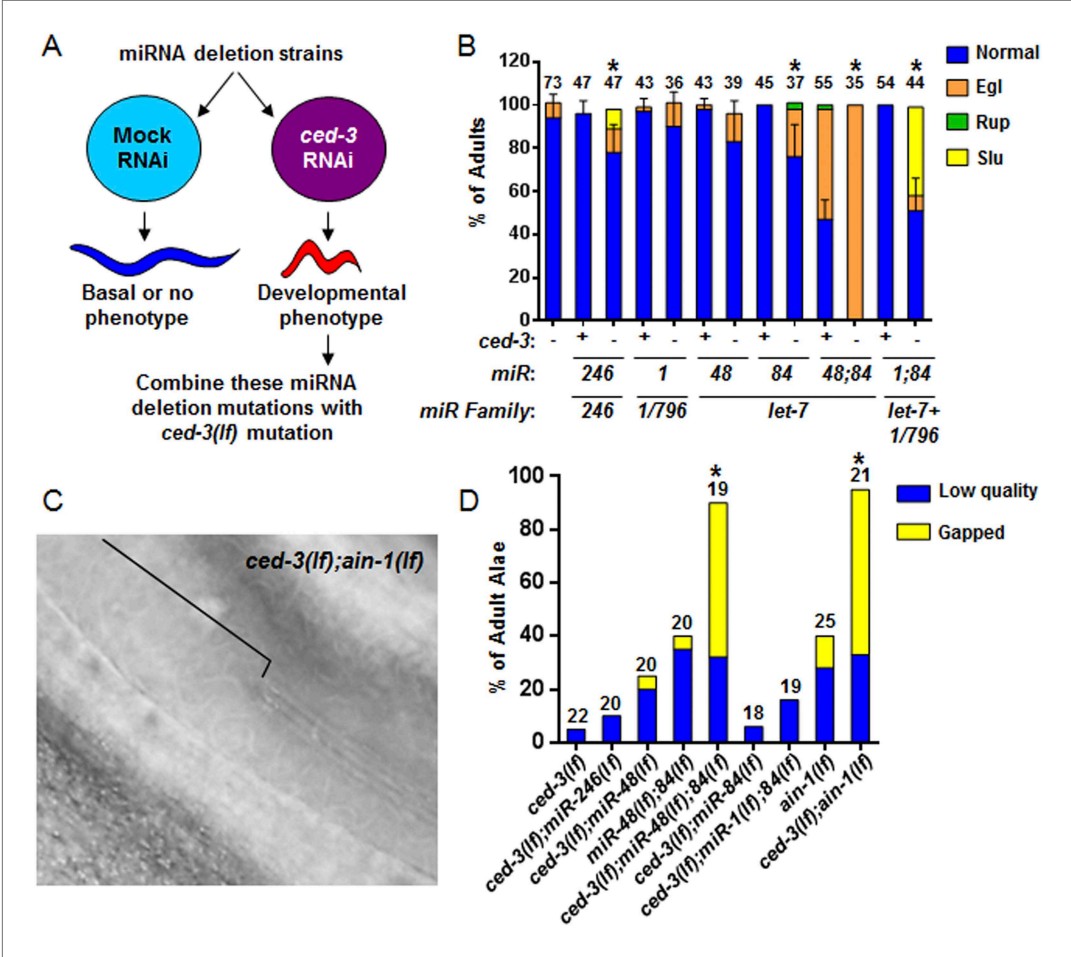

**Figure 5**. Identification of specific miRNAs that cooperate with *ced-3* caspase to regulate development. (**A**) Diagram for screening miRNA deletion mutants (listed in ***Supplementary file 4***) when fed mock or *ced-3* RNAi to identify overt developmental phenotypes when *ced-3* was depleted. *let-7(lf)* and *lin-4(lf)* mutants were excluded due to significant defects alone. (**B**) miRNA deletion(s) [indicated by the miR number(s)] identified in (**A**) were combined with *ced-3(lf)*. '+' and '−' indicate wild-type and *ced-3(null)*, respectively. Phenotypes including egg-laying defect (Egl), ruptured vulva (Rup), and sluggish movement (Slu) were quantified. Mean values ± SD for percent normal (p < 0.05, *when compared to *ced-3(lf)* and the relevant miRNA deletion(s) alone, Fisher's Exact test comparing the distributions of normal and abnormal animals). (**C** and **D**) *ced-3(lf)* enhances adult-specific alae defects including low quality (thin and rough) and gapped alae [bracket in (**C**) near the mid-body shows a gap]. Percent of adults with alae defects (p < 0.001, *compared to the relevant single or double mutants, Chi-square test comparing the distributions of adult alae phenotypes).

The following source data is available for figure 5:

**Source data 1**. Source data quantifying genetic interactions between miRNA mutants and *ced-3*.

proteolytic cleavage, which is consistent with our observation that LIN-14, LIN-28, and DISL-2 contain multiple consensus CED-3 cleavage sites that consist of a tetra-peptide sequence usually ending in an aspartic acid residue (*Xue et al., 1996*). To test this hypothesis, we performed an in vitro CED-3 cleavage assay as previously described (*Xue et al., 1996*). We found that the DIS3L2 ribonuclease homolog, DISL-2, was robustly cleaved by the CED-3 caspase while LIN-14 and LIN-28 were partially cleaved (*Figure 7A*). The multiple cleavage products generated by CED-3 cleavage of DISL-2 (*Figure 7A,B*) suggest a clear role for CED-3-mediated inactivation of this target protein. We further tested the specificity of the partial LIN-28 cleavage by CED-3 and found that it was completely blocked by addition of the caspase-specific-inhibitor zDEVD-fmk (*Figure 7C*). We then determined the

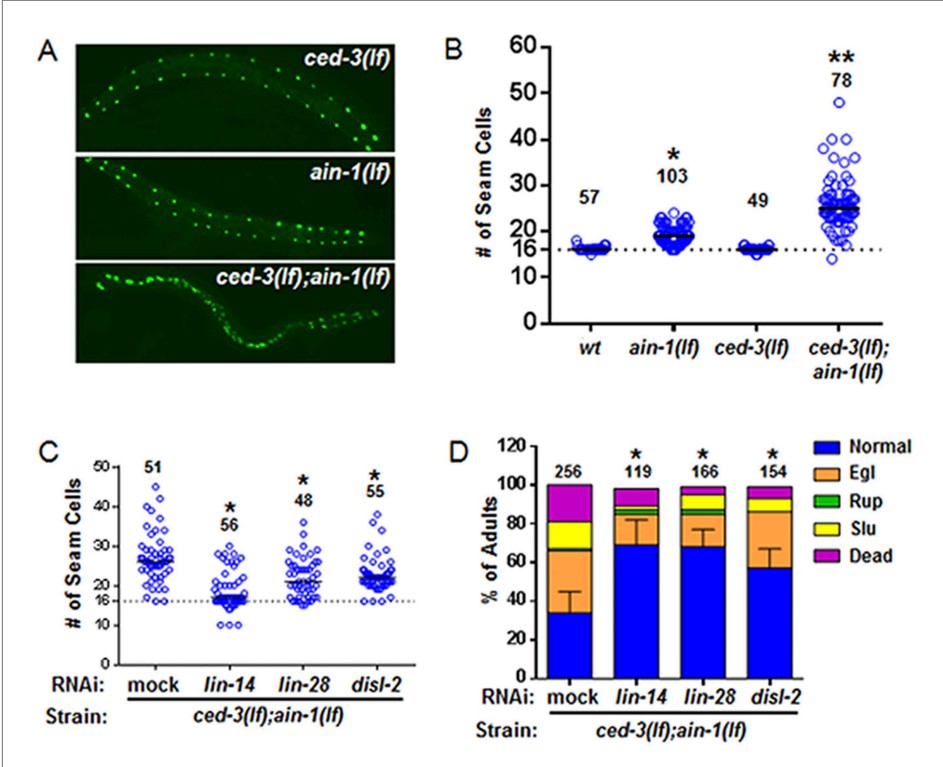

**Figure 6**. *ced-3* may act upstream of multiple conserved pluripotent factors to affect differentiation of stem cell-like seam cells. (**A** and **B**) Pseudocolored GFP from DIC images of a seam cell reporter and dot plot quantitation. The tick line depicts 16 seam cells that are normally found in wild-type animals. Black bars indicate the median values for each strain (p < 0.0001, *compared to wt, **compared to single mutants, Mann–Whitney test). (**C**) Effect of RNAi treatment beginning at L2 on the seam-cell-number phenotype of the *ced-3(lf);ain-1(lf)* double mutant (p < 0.0001, *compared to mock RNAi, Mann–Whitney test). *C. elegans disl-2* is homologous to mammalian Dis3l2 (*Figure 6—figure supplement 1*). (**D**) Effect of the same RNAi on the *ced-3(lf);ain-1(lf)* double mutant defects. Mean values ± SD for percent normal [p < 0.0001, *compared to mock RNAi, Fisher's Exact test comparing the distributions of normal and abnormal animals (see 'Materials and methods' for statistical rationale)].

The following source data and figure supplements are available for figure 6:

**Source data 1**. Source data quantifying temporal cell fate patterning and other phenotypes.

**Figure supplement 1**. Protein sequence alignment of human DIS3L2 and *C. elegans* DISL-2.

**Figure supplement 2**. Additional analyses of seam cells for the *ced-3(lf);ain-1(lf)* double mutant.

**Figure supplement 3**. *ced-3(lf)* mutants enhance *lin-66(RNAi)* ruptured vulva phenotype.

proteolytic cleavage site for LIN-28 by mutagenesis and identified the CED-3-specific recognition sequence (*Figure 7D* and *Figure 7—figure supplement 1*). Numerous possible cleavage sites were found for LIN-14 and DISL-2 but were not pursued further (*Figure 7—figure supplement 2*). The identified sequence DVVD fits the canonical CED-3 recognition motif (DxxD) (*Xue et al., 1996*) and mutating the second aspartic acid residue to an alanine (D31A in *Figure 7D*) entirely eliminated CED-3 cleavage. CED-3 proteolysis of LIN-28A generates an N-terminal asparagine in the remaining protein (*Figure 7E*). Asparagine is known to function generally as a destabilizing residue at the N-terminus of eukaryotic proteins resulting in proteasomal degradation in a phenomenon termed the N-end rule (*Sriram et al., 2011*).

## CED-3 impact on LIN-28 turnover in vivo

To examine CED-3-mediated turnover of the LIN-28 protein in vivo, we generated a polyclonal antibody against a C-terminal peptide in LIN-28 that recognizes both LIN-28 isoforms reported previously

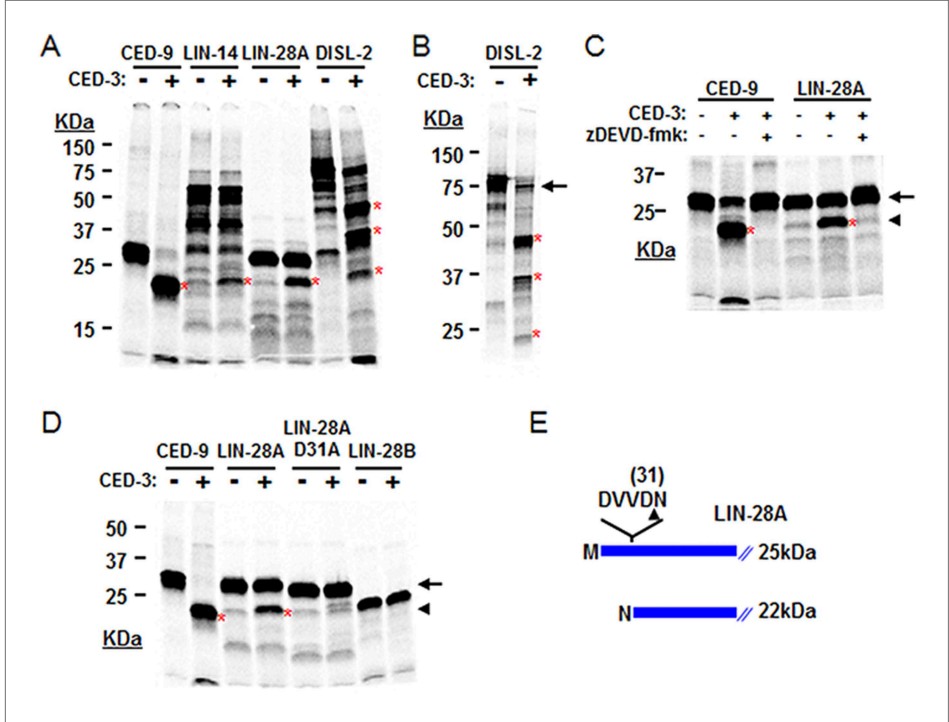

**Figure 7**. CED-3 cleavage of LIN-14, LIN-28, and DISL-2 (DIS3L2) in vitro. (**A**) Established in vitro CED-3 cleavage assay (*Xue et al., 1996*) of ³⁵S-labeled proteins. CED-9 served as a positive control throughout. Red asterisks indicate cleavage products (same in **B–D**). (**B**) Result from a longer-run gel showing near quantitative cleavage of full-length DISL-2 (arrow indicates the full-length protein). (**C**) In vitro cleavage assay with the zDEVD-fmk caspase-specific irreversible inhibitor (*Rickers et al., 1998*). The arrow and arrowhead (and red asterisks) indicate the full-length protein and a predominant CED-3 cleavage product, respectively. (**D**) Effect of the D31A mutation on CED-3 cleavage (for other mutants see *Figure 7—figure supplement 1*). (**E**) Diagram showing the position and consequence of LIN-28A cleavage by CED-3 in vitro (22 kDa with an N-terminal asparagine). Each panel was performed as an independent experiment.

The following figure supplements are available for figure 7:

**Figure supplement 1**. Mutagenesis of LIN-28A to identify the CED-3 cleavage site.

**Figure supplement 2**. Identification of possible CED-3 cleavage sites in LIN-14 and DISL-2.

(*Seggerson et al., 2002*) (*Figure 8—figure supplement 1A,B*). We found that the dynamic decrease in LIN-28 abundance during L2–L4 stages was similarly delayed by two different *ced-3(lf)* mutations (*Figure 8A* and quantitation shown in *Figure 8—figure supplement 1C*). At late L4 (48 hr in *Figure 8A*), LIN-28 was almost completely absent in both wild type and *ced-3(lf)* mutants, indicating the role of general, non-CED-3-mediated, proteolysis during late larval stages. Interestingly, the 22-kDa cleavage product observed in the in vitro assay (*Figure 7D,E*) was not observable in vivo (*Figure 8A*), consistent with the idea that the cleavage product with an asparagine at its N-terminus was possibly degraded by an additional proteolytic process. It is possible that the delayed down-regulation of LIN-28 seen in *Figure 8A* is the consequence of the slower post-embryonic growth rate observed for *ced-3(lf)* mutants (*Figure 4*). To address this question, we first used a LIN-28::GFP trans-genic strain previously shown to have functional LIN-28 activity (*Moss et al., 1997*) to monitor stage-matched L3 larvae with or without a *ced-3(lf)* mutation by DIC microscopy. We observed that the *ced-3(lf)* mutation delayed the proper down-regulation of the LIN-28::GFP reporter at L3 in the hypodermis (*Figure 8B–D* and *Figure 8—figure supplement 2A–C*). We also found that down-regulation of LIN-28::GFP expression was delayed in neuronal cells in the head (*Figure 8—figure supplement 2D,E*). These findings support the hypothesis for the delayed down-regulation of LIN-28

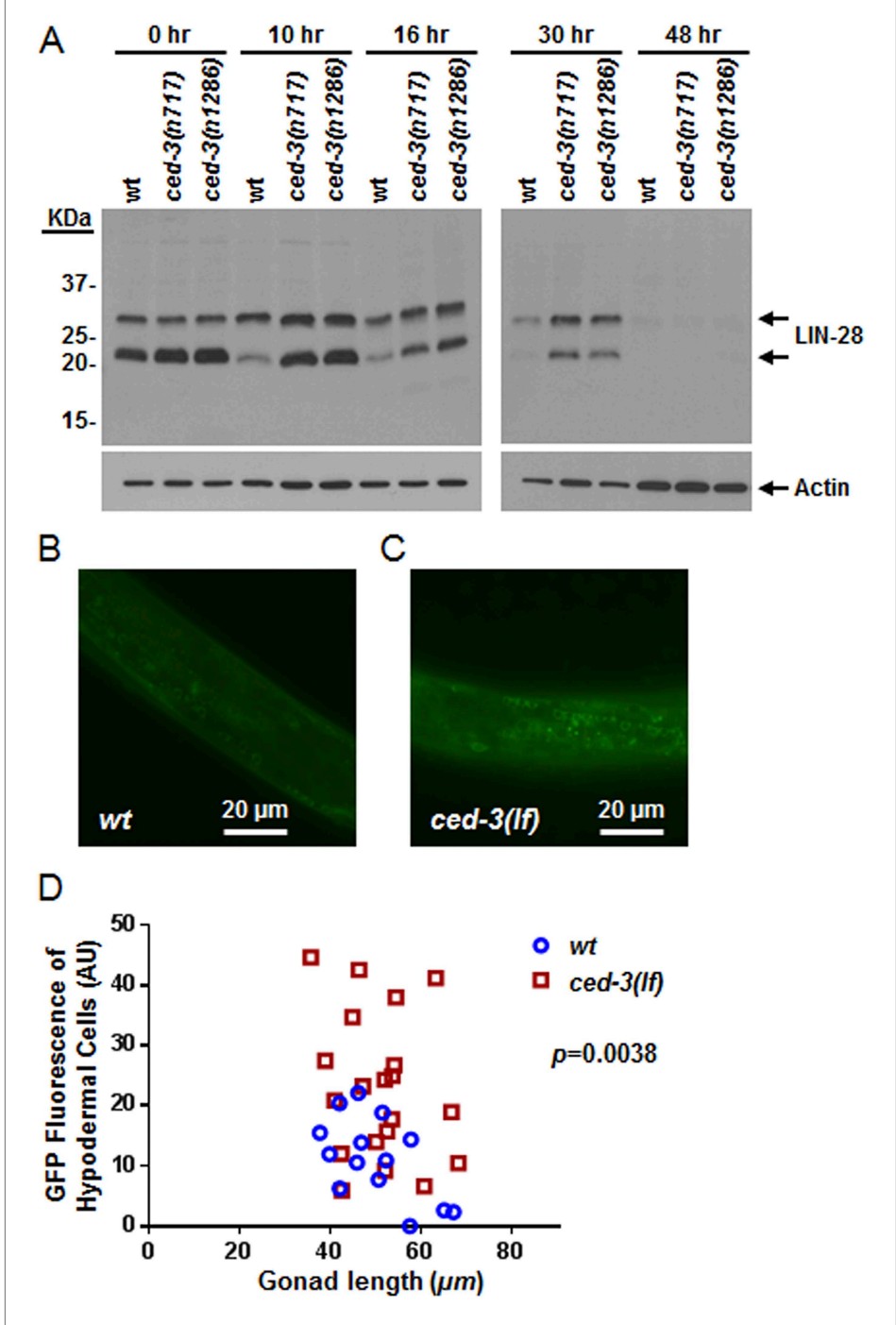

**Figure 8**. In vivo confirmation that CED-3 caspase negatively regulates LIN-28 expression in late larval stages.
(**A**) Western blot with the LIN-28 antibody we developed (validation shown in *Figure 8—figure supplement 1*) to
see the effects of *ced-3(lf)* mutations on LIN-28 protein expression during developmental transitions. Notice that
the cleavage product of the larger isoform of LIN-28 observed in the in vitro assay (*Figure 7*) is not detectable in
the in vivo analysis, suggesting that the cleavage product, which has an Asn instead of Met as the N-terminal end
residue, may potentially be sensitive to the N-end rule proteasomal degradation pathway. The pattern and timing
of LIN-28 expression and downregulation we show here for wt are similar to previous findings (*Seggerson et al.,
2002*; *Morita and Han, 2006*) and two independent *ced-3(lf)* mutant strains are shown here (quantitation is
shown in *Figure 8—figure supplement 1C*). (**B–C**) Pseudocolored GFP from DIC images of L3 larvae near the
mid-body. Also see DIC images of these same animals without GFP illumination (similar length gonads shown in
*Figure 8. Continued on next page*

*Figure 8. Continued*

*Figure 8—figure supplement 2A,B*) and test of similar staging (*Figure 8—figure supplement 2C*). Size bars are indicated. 'wt' indicates the *lin-28(+)::gfp* integrated transgene alone previously shown to be functional (*Moss et al., 1997*) and 'ced-3(lf)' indicates this same transgene combined with a *ced-3(lf)* mutation. (**D**) Quantitation of the LIN-28::GFP expression between the strains within the L3 stage. (p = 0.0038, significant compared to *wt*, Mann–Whitney test comparing the integrated intensity of LIN-28::GFP hypodermal expression in L3 larvae). Persistent expression of LIN-28::GFP in head cells (*Figure 8—figure supplement 2D,E*).

The following source data and figure supplements are available for figure 8:

**Source data 1**. Source data quantifying effects of *ced-3(lf)* on LIN-28::GFP expression.

**Figure supplement 1**. Validation of our newly generated LIN-28 antibody and quantitation of Western blot data.

**Figure supplement 2**. Larval staging and persistent LIN-28::GFP expression.

by *ced-3(lf)*. The difference in magnitude between the Western blot results and the number of fluorescent cells seen by DIC microscopy may suggest that the observed fluorescence levels do not linearly reflect the protein levels and that the two methods may have different dynamic ranges.

We then further addressed the question by testing the physiological impact of the *lin-28(D31A)* mutation. Specifically, we made the point mutation in the previously published *lin-28(+)::gfp* fusion protein (*Moss et al., 1997*). To ensure that the LIN-28(D31A) mutation did not disrupt the global function of the protein, we tested its ability to overcome the highly penetrant protruding vulva (Pvl) phenotype in *lin-28(n719,lf)* animals and found that it was able to rescue the Pvl phenotype (*Figure 9—figure supplement 1*). Following integration and outcrossing, we found the copy number of the *lin-28(D31A)::gfp* transgene to be slightly lower than that of the non-mutated *lin-28(+)::gfp* transgene (*Figure 9—figure supplement 2*). We then examined the developmental profile and found that the *lin-28(D31A)::gfp* transgene alone caused a delay in larval development similar to that caused by the combination of the *lin-28(+)::gfp* transgene with *ced-3(lf)* (*Figure 9A*). Western blot analysis showed that the *lin-28(D31A)::gfp* integration had less basal expression than the non-mutated *lin-28(+)::gfp* integration, consistent with the lower copy number estimate. We observed a quantifiable difference in the down-regulation of the *lin-28(D31A)::gfp* transgene compared to the *lin-28(+)::gfp* transgene (*Figure 9B,C*). This finding provides evidence that a failure in CED-3 cleavage of LIN-28 leads to slower degradation of LIN-28 and is one of the causes of slower development, since the D31A point mutation alone resulted in both a slower growth rate (*Figure 9A*) and delayed LIN-28 down-regulation (*Figure 9B,C*). Additionally, in this Western blot (*Figure 9B–C*), down-regulation of the wild-type LIN-28 transgene in *ced-3(lf)* worms seems to be delayed more than LIN-28(D31A) in wild-type worms. Such a difference could be due to roles of CED-3 on other targets such as LIN-14 and DISL-2, which is also expected to contribute to the larval developmental defect in *ced-3(lf)* (*Figure 9A*).

Examination of adult-specific alae is a sensitive physiological readout that should overcome any limitations of monitoring delays in the down-regulation of LIN-28 expression levels since scoring adult alae ensures stage-matching and accounts for any perdurance. To further test the functional outcome of both the LIN-28(D31A) transgene and the LIN-28(+) transgene combined with *ced-3(lf)*, we examined the adult-specific alae and found significant defects including low quality and gapped alae (*Figure 9D* and *Figure 9—figure supplement 3*). This is consistent with the data described above that *ced-3(lf)* enhances adult-specific alae defect of let-7-family miRNA mutants and *ain-1(lf)* (*Figure 5C–D*). We should note that the original report of the LIN-28(+) transgene indicated that some of the adults were observed to have gapped alae (*Moss et al., 1997*). Though we did observe rough and very thin sections of alae for this strain (scored as low quality alae), we did not observe any gapped adult alae. This subtle difference is likely explained by a different threshold since we scored alae using a sensitive camera (See 'Materials and methods'). Nonetheless, the relative enhancement of *ced-3(lf)* with this transgene is quite obvious and similar to that of the caspase-cleavage resistant LIN-28(D31A) point mutant transgene (*Figure 9D*). Altogether, our data support a causal role for CED-3 cleavage of LIN-28 in the regulation of temporal cell fate patterning. CED-3 appears to facilitate the stereotypical transition of LIN-28 to enhance the robustness of the L2 to L3 developmental transition.

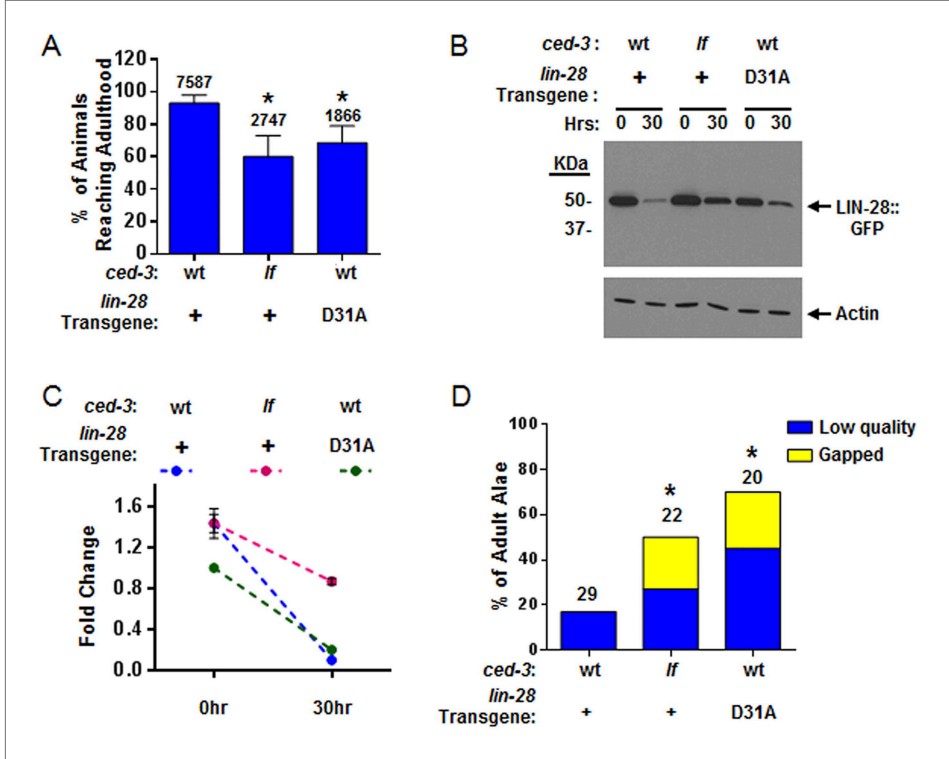

**Figure 9**. CED-3 caspase represses LIN-28 in vivo to ensure proper temporal cell fate patterning regulation. (**A**) Effects of disrupting CED-3 activity on LIN-28 in vivo on the rate of post-embryonic growth. Percent of animals reaching adulthood at 96 hr after hatching is shown. '+' indicates the *lin-28(+)::gfp* integrated transgene described in *Figure 8B–C*. D31A indicates a transgene integration with the CED-3-cleavage-resistant D31A point mutation in the first exon of LIN-28 but is otherwise identical to the original (+) transgene. Test of *lin-28(lf)* rescue (*Figure 9—figure supplement 1*) and copy number of the transgenes (*Figure 9—figure supplement 2*). Mean values ± SD (p < 0.0001, *compared to wt;(+), Fisher's Exact test comparing the distributions of adult to larval-stage animals at this time). (**B–C**) Western blot for the LIN-28::GFP transgenes described in (**A**) and quantitation from three independent Western blot experiments of the LIN-28::GFP transgenes [one Western blot shown in (**B**)]. Here, 1.0 was defined as the intensity of total LIN-28(D31A)::GFP at 0 hr normalized to actin. Both the *lin-28(+)* and the *ced-3(lf);lin-28(+)* strains and the 30 hr time point for all strains were compared to this value. Mean ± SEM for the two time points (dashed lines are used only to indicate the net change in relative expression for the three strains). (**D**) Disrupting CED-3 activity on LIN-28 enhances adult alae defects of the strains described in (**A**) (p < 0.01, *compared to wt;(+), Chi-square test comparing the distributions of adult alae phenotypes). *Figure 9—figure supplement 3* shows examples of the adult alae phenotypes for these three strains. Data for increased expression of LIN-14 in *ced-3(lf)* mutants at the first larval stage is shown in *Figure 9—figure supplement 4*.

The following source data and figure supplements are available for figure 9:

**Source data 1**. Source data quantifying effects of *ced-3(lf)* and LIN-28(D31A) mutation on protein levels and developmental phenotypes.

**Figure supplement 1**. Test for *lin-28(D31A)* function in overcoming the *lin-28(n719,lf)* protruding vulva defect.

**Figure supplement 2**. Transgene copy number determination.

**Figure supplement 3**. Loss of *ced-3* function or mutating the CED-3 cleavage site of LIN-28 enhances adult alae defects by a multi-copy *lin-28* transgene.

**Figure supplement 4**. LIN-14 protein levels are increased in *ced-3(lf)* mutants at the first larval stage.

Consistent with LIN-14 being modestly cleaved by CED-3 in vitro (*Figure 7A*), we found that the LIN-14::GFP level was modestly increased in *ced-3(lf)* mutants in vivo at the L1 stage (*Figure 9—figure supplement 4*). This result may not be explained by slower growth rate since these animals were obtained as synchronous L1s without food. Our attempts to monitor DISL-2 protein levels including developing an antibody to endogenous DISL-2 were impeded by technical difficulties. Moreover, N- and C-terminal GFP fusions to DISL-2 had exceedingly low levels of expression beyond detection by common methods suggesting that DISL-2 protein levels are kept exquisitely low for physiological significance.

Therefore, our in vitro and in vivo data show that developmental timing regulators are proteolytic targets of the CED-3 caspase, likely resulting in their inactivation. This role of CED-3 cleavage is in contrast to known apoptotic functions of CED-3 caspase activity in two major aspects: CED-3 inactivates its targets rather than activates them as in its apoptotic function (*Conradt and Xue, 2005*); and it acts with other regulatory systems, including miRNAs and possibly the N-end rule proteasomal system, to maintain robust developmental functions.

## Discussion

### Role of non-apoptotic CED-3 activity in enhancing the robustness of dynamic changes in gene expression for development

We report the discovery of a new gene expression regulatory mechanism whereby a non-apoptotic activity of the CED-3 caspase functions to inactivate and repress the expression of key developmental regulators, significantly contributing to the robustness of gene expression dynamics and animal development (*Figure 10A*). Consistent with this, a previous report showed that CED-3 is capable of cleaving more than 22 *C. elegans* proteins in an in vitro proteomics survey (*Taylor et al., 2007*) and two recent genetics-based findings showed that *ced-3* may play important roles in neural regeneration (*Pinan-Lucarre et al., 2012*) and aging (*Yee et al., 2014*). Second, the described CED-3 function in repressing gene expression is likely in contrast to the role of CED-3 in promoting apoptosis through activation of protein targets by cleavage at specific sites (*Nakagawa et al., 2010*; *Chen et al., 2013*). Here, the CED-3 cleavage alone may already destroy the target protein activity. Additionally, the cleavage products may be further degraded by other degradation systems notably via N-terminal destabilizing residues which may make the target more susceptible to additional degradation mechanisms, such as proteasomal degradation (*Sriram et al., 2011*) (*Figure 10B*). We hypothesize that this function operates continually during development to facilitate rapid turnover of these regulatory proteins at the post-translational level and in cooperation with other regulatory mechanisms (*Figure 10—figure supplement 1*). We should note that it is curious that only the LIN-28A isoform was found to be cleaved by CED-3 in vitro yet expression of both LIN-28A and LIN-28B isoforms was altered by *ced-3(lf)* in vivo. This may imply that *ced-3* has potential indirect effects on other factors within the heterochronic pathway that could alter LIN-28 isoform expression but further experiments are required to satisfactorily explain this.

We find that the altered LIN-28 expression levels in a *ced-3(lf)* background or with the caspase-cleavage resistant mutant [LIN-28(D31A)] in a *ced-3(wt)* background, are subtle compared to previous findings regarding a *lin-28(gf)* transgene with deleted *lin-4* and *let-7* miRNA-binding sites in the 3′ UTR (*Moss et al., 1997*). Consistent with this subtlety, *ced-3(lf)* alone displays essentially no defect in seam cell numbers (*Figure 6*). The physiological effect of this subtle regulation is clearly seen in seam cell temporal patterning when miRNA function is compromised in the *ain-1(lf)* mutant background. This prominent enhancement indicates that *ced-3* has an important role in supporting the robustness of the larval transitions. Based on the pleiotropic phenotypes associated with *ced-3(lf);ain-1(lf)*, such roles may potentially extend to a broad range of cellular processes.

### Cooperative gene regulation revealed by our genome-scale screen

Previous studies using model organisms, including our own, have indicated that genetic redundancy by structurally unrelated genes is commonly associated with genes with regulatory functions (*Ferguson et al., 1987*; *Fay et al., 2002*; *Suzuki and Han, 2006*; *Costanzo et al., 2011*). Asking the same question for the global miRISC function, our screen, by identifying 118 previously unknown miRISC interactors, thus identified new roles for miRISC in normal developmental processes that are otherwise masked by redundancy and/or pleiotropy, as well as identifying other regulatory mechanisms that collaborate with miRNAs. Examples we found for the latter in this study

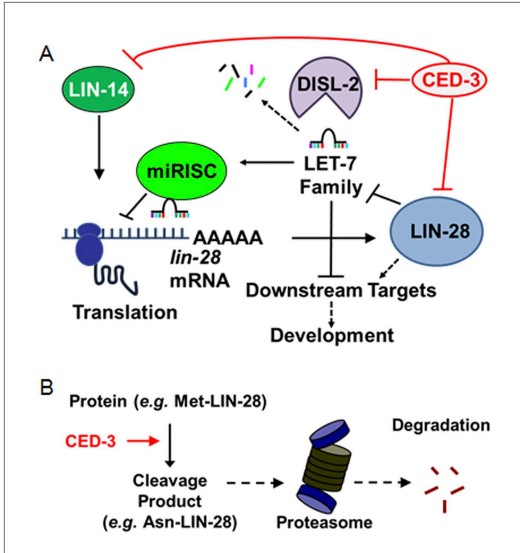

**Figure 10**. Model for CED-3 function in temporal cell fate patterning regulation. (**A**) Model of CED-3 collaborating with miRNAs to repress the expression of LIN-14, LIN-28, and DISL-2. Red blocks indicate the new findings. For simplicity, many other factors in the pathway were not included here including additional regulators associated with this pathway that were also identified in our genomic enhancer screen (see *Figure 10—figure supplement 1*). (**B**) Hypothetical model for the biochemical role of CED-3 cleavage in protein turnover during development whereby a new N-terminus is generated which could potentially destabilize the protein according to the N-end rule (see text).

The following figure supplement is available for figure 10:

**Figure supplement 1**. A more detailed genetic model for roles of CED-3 caspase in regulating the heterochronic pathway and for showing other genes from our genomic screen in this pathway.

include genes encoding the POU-homeodomain protein (*ceh-18*, *Figure 2—figure supplement 1*), the histone acetyltransferase (*pcaf-1*), the ras-related GTPase homolog (*ral-1*), the homeodomain transcription factor (*unc-39*), and the cell-killing *ced-3* caspase (the majority of this study) (and others listed in *Supplementary file 2*). However, the interactions identified in this study most likely reflect only a small portion of miRNA functions because screening for obvious developmental defects under well-fed conditions only permitted us to identify limited physiological functions. Applying various assays, including behavioural assays or animals under various growth or stress conditions, is expected to identify many more miRNA functions. Furthermore, although feeding RNAi has important advantages for such a screen, it is not effective for many genes especially for genes functioning in certain tissues such as neurons. Therefore, genetic screens or analyses under sensitized backgrounds will continue to play a major role in identifying miRNA functions.

## Materials and methods

### *C. elegans* strains
See *Supplementary file 4* for the list of all strains used in this study.

### Rationale of phenotypes scored in this screen
In this screen, we wanted to identify genetic pathways that may redundantly cooperate with the miRISC in development. Since the loss of most miRISC function resulted in highly pleiotropic phenotypes (*Zhang et al., 2009*), we chose to score multiple obvious phenotypes (defined in *Supplementary file 1*).

### Genome-wide, double blind RNAi screen
The ORFeome RNAi feeding library (*Rual et al., 2004*) was screened using a 96-well liquid culture format in the double blind. Here, double blind means that no identities for interactors were revealed to anyone setting up the plates, anyone phenotyping the plates, nor anyone processing the scored data until all candidate interactors were confirmed in a secondary screen performed in quadruplicate (see below). Similar to a previously reported method (*Lehner et al., 2006*), a 2 day set up for each screening session was employed (*Figure 1—figure supplement 1A*).

For each scoring session, *rrf-3(pk1426,lf), ain-1(ku322,lf);rrf-3(pk1426,lf)*, and *ain-2(tm2432,lf);rrf-3(pk1426,lf)* were each fed with mock, *ain-1*, and *ain-2* RNAi cultures in parallel which served as the experimental controls. These controls were set up in 4 sets of triplicate (n = 12 total for each). We identified potential interactors whenever *ain-1(ku322);rrf-3(pk1426)* or *ain-2(tm2432);rrf-3(pk1426)* showed a significant defect (*Figure 1—figure supplement 1A*). All candidates were then retested in quadruplicate liquid format. Any gene showing effect in three or more replicates was considered a *bona fide* interactor by RNAi and their identities were then revealed and confirmed by sequence analysis. Multiple interactors were confirmed by testing the corresponding mutant strains when treated with *ain-1* or *ain-2* RNAi (*Supplementary file 3*).

## Statistical analyses

Before any statistical analyses were made, all relevant data sets were first tested for normality using the D'Agostino-Pearson omnibus test. This test also informed us for sufficient sample sizes. We analyzed our results in the following ways: (1) the Mann–Whitney test was used for pair-wise comparisons, (2) Chi-square analysis was used to compare distributions of categorical data, and (3) Fisher's Exact test was used to analyze cases where two categories were most important between two strains (e.g., the frequency of normal animals to the pooled frequency of all abnormal animals in each of the tissue-specific rescues or in the RNAi suppression test). Use of Fisher's Exact test in such cases prevented outcomes where Chi-square analysis of the same data may identify a rescue as significant only because the abnormal phenotypic categories had changed in distribution relative to the unrescued mutant, but where the fraction of normal animals was not improved. p values and statistical tests were reported throughout the study. Statistics source data have been provided.

## RNAi treatment by feeding on solid agar

Similar to the liquid culture format, positive and negative controls were run in parallel to ensure effectiveness of the culturing conditions. RNAi cultures and plates were prepared as previously described (*Fraser et al., 2000*; *Timmons et al., 2001*) with 100 µg/ml ampicillin. Depending on the experiment, strains were added to RNAi plates in one of the following ways: (1) bleached strains were synchronised in M9 for 36 hr at 20°C, counted, then added to plates or (2) either eggs, L2 stage animals, or L4 stage animals were carefully added to lawns.

## Counting percent of eggs hatched after RNAi treatment

Gravid young adult *ain-1(lf)* hermaphrodites fed either mock or *ced-3* RNAi since hatching were transferred to a new RNAi plate and allowed to lay eggs for 4–5 hr. The young adults were removed and the eggs laid were counted. After 40 hr at 20°C, unhatched eggs and larvae were scored. 64 hr after removing young adults, very few additional larvae were observed for *ain-1(lf)* animals treated with *ced-3* RNAi. Data are from eight independent trials.

## Assay for the rates of post-embryonic development

Synchronous L1 stage animals were added to normal food (OP50 bacteria) (150–200 worms per trial) and incubated at 20°C. Animals were scored for developmental stages every 24 hr thereafter. Data are from three to five independent trials.

## Tissue-specific rescue of *ain-1(lf)*

The *ain-1(lf)* defects were rescued using a fragment of genomic DNA containing *ain-1* sequence and native promoter (*Ding et al., 2005*), an *ain-1* sequence driven by a *dpy-5* (hypodermal-specific [*Thacker et al., 2006*]) or a *ges-1* (gut-specific [*Egan et al., 1995*]) promoter, and genome-integrated constructs with tissue-specific promoters driving *ain-2* expression which has previously been shown by our lab to rescue *ain-1(lf)* in those tissues (*Zhang et al., 2011*; *Kudlow et al., 2012*; *Than et al., 2013*). Data are from four to ten independent trials.

## Enhancer screen for *miRNA* deletion mutants with *ced-3* RNAi

In the blind, all available miRNA deletion mutants were tested for enhancer phenotypes with *ced-3* using the RNAi feeding method on solid agar. The *let-7(lf)* and *lin-4(lf)* mutants were excluded since they are very sick. One person picked 10 eggs or 2 L4 stage animals onto mock or *ced-3* RNAi in replicates according to a key that was kept confidential. 4 to 5 days later, another person then examined the plates for phenotypes (defined in *Supplementary file 1*). All mutants showing an RNAi phenotype were revealed for identity and then crossed with *ced-3(lf)* mutants in single or in combination and tested for enhancer phenotypes.

## Apoptotic assay and rationale

We employed a published assay to identify subtle apoptotic enhancers using a reporter line: *ced-3(n2427*, reduction of function);*nIs106 [lin-11::GFP + lin-15(+)X]* (*Reddien et al., 2007*). The *ced-3(n717);nIs106* strain served as the positive control for complete loss of *ced-3* function, and the *mcd-1(n3376);ced-3(n2427,rf);nIs106* strain was the positive control for enhanced ablation of programmed cell death comparable to the strong *ced-3(n717)* loss of function allele for accumulation of P9-11.aap cells, consistent with the previous findings (*Reddien et al., 2007*). Young adults of all

strains were scored in the blind for the number of GFP-positive undead P9-11.aap ventral cord cells. Three independent lines of *ced-3(n2427,rf);ain-1(lf);nIs106* were scored in the blind (data for these three lines were combined in *Figure 3B*).

### L1 stage cell-corpse assay

This standard method was done as previously described (*Ledwich et al., 2000*). The *ced-1(e1735)* mutation was used to enhance visualization of corpses. DIC optics were used to count the head corpses.

### 3′ UTR miRNA seed site prediction

These predictions were all made by TargetScan 6.2 release June 2012 (*Jan et al., 2011*; *Lewis et al., 2005*).

### Scoring adult-specific alae

Adult alae were scored using DIC optics (Zeiss Axioplan 2, Thornwood, NY) at 630× magnification (*Ding et al., 2005*; *Zhang et al., 2007*, *2009*). One side of each non-roller adult was scored (the side facing up). All roller phenotype animals (the three LIN-28::GFP transgenic lines) were scored in the same way such that all alae that could be viewed were assessed for gaps and quality. Each animal was scored as either normal, low quality alae (very thin and rough sections) or gapped alae (discontinuous alae). Animals with both low quality and gapped alae were counted as only gapped alae so that each animal was represented only once. Any thin region of alae that appeared as a gap through the oculars was imaged by the camera (Zeiss Axiocam MRm) and evaluated on a large screen. Only alae observed as truly discontinuous by aid of the camera were scored as gapped. This method was applied equally to all strains throughout the study.

### Seam cell counting method

All seam cell lines were counted on a fluorescent microscope with DIC optics (Zeiss Axioplan 2) at 110× and 630× magnification (*Zhang et al., 2009*) at the L1, L3, or L4 stage. To prevent over-representation of our sample size, we reported only one side of each animal. We randomly chose to report the top or the left side of the animal, depending on the orientation in the microscopy field. We followed this convention for the single mutants as well. Therefore, one dot corresponds to one side of one animal and each animal is plotted only once (*Figure 6A–C* and *Figure 6—figure supplement 2*). Data are from five independent trials.

### RNAi suppression test

We hypothesized that loss of both *ain-1* and *ced-3* resulted in the upregulation of LIN-14, LIN-28, and DISL-2. These factors are normally expressed at high levels beginning in late embryonic development and down-regulated toward the end of the second larval stage. We therefore decided to begin RNAi treatment of *ced-3(lf);ain-1(lf)* animals at the second larval stage and score for phenotypes 48–54 hr later. Animals were considered normal if they were only mildly-to-moderately egg-laying defective and capable of normal motility. Data are from three to six independent trials.

### CED-3 in vitro cleavage assay

The LIN-14, LIN-28, and DISL-2 coding sequence templates for in vitro synthesis were each generated first by reverse transcription from mixed stage WT (N2) *C. elegans* total RNA and then PCR amplified before subcloning into pTNT vector (Promega, Madison WI). The primer sequences are as follows (Restrictions sites indicated in bold-type, start codons underlined in FWD primers): *lin-14* FWD, att**acgcgt**ACCATGGCTATGGATCTGCCTGGAACGTCTTCGAAC; REV, att**ggtacc**CTATTGTGGAC-CTTGAAGAGGAGGAG; *lin-28* FWD, att**acgcgt**ACCATGGCTATGTCGACGGTAGTATCGGAGGGA; REV, att**ggtacc**CTCAGTGTCTAGATGATTCTATTCATC; *disl-2* FWD, att**acgcgt**ACCATGGCTATGTCAGCAGTT-GAAAGTCCCGTT; REV, att**ggtacc**CTACTGAAGAATTGTTGAGCCCGTTTC. Point mutations were generated using Quick Change II kit (Agilent Technologies, Santa Clara, CA). All constructs were sequence-verified. As previously published (*Xue et al., 1996*), cleavage substrates were freshly synthesised with L-$^{35}$S-Methionine in vitro and used immediately. For caspase inhibitor reactions, zDEVD-fmk caspase-specific inhibitor (ApexBio, Houston, TX) or DMSO was added. All cleavage reactions were incubated at 30°C in a thermocycler with heated lid for up to 6 hr. Each panel shown in *Figure 7* was performed independently with freshly synthesized L-$^{35}$S-labeled substrates and independent cleavage reactions for each experiment.

## LIN-28 antibody and Western blot

Antibody against a LIN-28 C-terminal peptide (RKHRPEQVAAEEAEA) was produced by Spring Valley Laboratories (Sykesville, MD) using rabbit as the host and purified using a peptide column. Validation of the specificity of the antibody is shown in *Figure 8—figure supplement 1A,B*. Synchronous L1 stage animals were added to normal food (OP50 bacteria) and incubated at 20°C then collected at the indicated hours with food. For each time-point, equivalent protein input from wt, *ced-3(n717)*, and *ced-3(n1286)* staged animal lysates were resolved by SDS-PAGE and then detected by Western blot using the anti-LIN-28 antibody. Actin was used as loading control (Anti-Actin antibody, A2066, Sigma–Aldrich, St. Louis, MO).

## Scoring LIN-28::GFP positive cells by DIC optics

Similarly sized L3 stage animals were picked on a non-fluorescent dissecting scope to blind the selection of animals. Prior to fluorescent illumination, gonad length was observed and measured to ensure animals were of comparable developmental stage (*Ambros and Horvitz, 1984*; *Moss et al., 1997*; *Abbott et al., 2005*). This method should provide a similar distribution of developmental sub-stages for both backgrounds within the L3 stage. No significant difference in gonad extension was found (*Figure 8—figure supplement 2A–C*). Gonad length was measured and recorded prior to GFP illumination to ensure no bias. All animals were illuminated for 5 s for each picture by DIC optics. Multiple planes through the animal were examined by one person to ensure all GFP positive cells were identified. Another person, who did not take the images, then used ImageJ to obtain integrated GFP intensity values which were reported relative to the gonad length to account for stage (*Figure 8B–D*) or counted the number of GFP positive head cells (*Figure 8—figure supplement 2D,E*). Data for all animals viewed by DIC were kept and reported. Data for the hypodermal and head cell expression assays come from two and three independent experiments, respectively.

## Acknowledgements

We thank R Horvitz, S Mitani, V Ambros, and the CGC (funded by NIH Office of Research Infrastructure Programs (P40 OD010440)) for strains; E Moss for materials; A Fire for vectors; V Ambros, W Wood, R Yi, S Park, M Kniazeva, M Cui, and Han lab members for discussions; and A Sewell for comments. Supported by the postdoctoral fellowship 121631-PF-12-088-01-RMC from the American Cancer Society (BPW), NIH grants 5R01GM047869 (MH), R01GM059083 (DX), and 2T32GM008759-11 (RZ). The authors of this study would like to declare no competing financial interests. The funding agencies had no role in study design, data collection, interpretation of the results, the decision to publish, or preparation of the manuscript.

## Additional information

### Funding

| Funder | Grant reference number | Author |
| --- | --- | --- |
| American Cancer Society | 121631-PF-12-088-01-RMC | Benjamin P Weaver |
| National Institute of General Medical Sciences | 2T32GM008759-11 | Rebecca Zabinsky |
| National Institute of General Medical Sciences | 5R01GM047869 | Min Han |
| National Institute of General Medical Sciences | R01GM059083 | Eui Seung Lee, Ding Xue |
| Howard Hughes Medical Institute | | Min Han |

The funders had no role in study design, data collection and interpretation, or the decision to submit the work for publication.

### Author contributions

BPW, Conception and design, Acquisition of data, Analysis and interpretation of data, Drafting or revising the article; RZ, YMW, Conception and design, Acquisition of data, Analysis and interpretation

of data; ESL, DX, Provided critical guidance, Contributed unpublished essential data or reagents; MH, Supervised the study

## Additional files

### Supplementary files
• Supplementary file 1. Definition of phenotypes scored in this study.

• Supplementary file 2. List of *ain-1* and *ain-2* genetic interactors identified in this study and their relevant phenotypes in brief. RNAi clones are listed alphabetically by gene name. Relevant phenotypes indicated for the given strains are defined in **Supplementary file 1**.

• Supplementary file 3. Phenotypes observed for reverse confirmation test with *ain-1* and *ain-2* RNAi. Effects indicated are relative to the given mutant strain phenotype on mock RNAi. These are results for one generation on the indicated RNAi and not with RNAi enhancing mutations.

• Supplementary file 4. List of *C. elegans* strains and relevant genotypes used in this study.

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
