## [Decision Letter]

Thank you for choosing to send your work entitled “Non-apoptotic CED-3 regulates LIN-28-LET-7 pluripotency pathway and other factors for robust development in *C. elegans*” for consideration at *eLife*. Your full submission has been evaluated by Detlef Weigel (Senior editor), a member of our Board of Reviewing Editors, and 2 peer reviewers, and the decision was reached after discussions between the reviewers.

We are in principle interested in the work, because it potentially reveals a new and exciting mechanism for CED-3 action. Unfortunately, the claims made in the article and abstract, which imply a direct effect of the CED-3 protease on the stability of Lin-28 present only one possible explanation of many for the observed data. Demonstrating such a direct effect would, however, be essential for eventual acceptance of the work. We would minimally expect direct *in vivo* observations of Lin28 protein stability in the appropriate ced3 genetic background in properly staged animals.

The main set of experiments that are not clear revolve around the interpretation of the LIN-28 time course in wild-type and ced-3 mutants. There is an important facet to the interpretation of these experiments that may be explained by the trivial notion that *ced-3* mutants develop more slowly than wild-type animals. The reviewers feel that this caveat could be addressed as you will see in the more detailed comments below. It would also be important to provide a clear definition of what is being phenotypically assayed.

Reviewer #1:

Summary: While the authors outline a nice genetic strategy to identify components that function with or in parallel to genes involved in general miRNA regulation and also clearly demonstrate that *ced-3* functions in this capacity and outside its role in cell death, the experiments that demonstrate LIN-28 regulatory defects in *ced-3* mutants are need expansion. The *in vitro* and genetic data are solid. Unfortunately, the phenotypes the statement that *ced-3* contributes to the post-translational regulation of various miRNA targets is not complete. The authors have all of the reagents to prove this assertion follow-up experiments are outlined below.

Genetic screen and strategy: The authors present the findings of a genome-wide RNAi screen designed to identify factors that work together or in parallel to essential components of the miRNA machinery that regulates developmental gene expression. The basis for this screen relies on the fact that *ain-1*, encoding the *C. elegans* miRISC component/GW Protein homolog, is non-essential (due to a redundancy with the *C. elegans ain-2* gene), displays a variety of pleiotropic phenotypes that can serve as the starting point for an enhancer screen. To identify genetic interactors, the authors systematically depleted each protein-coding gene individually by RNAi and scored for a wide range of general phenotypes. This resulted in the identification of 126 genes, that when depleted, result in a reproducibly synergistic enhancement of *ain-1* and *ain-2* mutant phenotypes. This list is composed of genes that function in a wide array of cellular and development functions and is interesting in itself with regard to the many functions for miRNAs in normal development.

Reviewer #2:

The manuscript by Weaver et al. presents the data from a carefully controlled genome-wide RNAi screen to search for genes whose depletion enhances phenotypes of animals compromised for miRISC function. This data set is likely to be of broad interest to the scientific community. A significant and somewhat surprising finding from this screen is the discovery that the CED-3 caspase has non-apoptotic roles in multiple developmental processes. Other recent studies have implicated *ced-3* in similar such functions, but did not pursue *ced-3* involvement to the level of protein targets. One key aspect of this paper is the identification of candidate *ced-3* target proteins with bearing on the synthetic phenotype. However, the experiments to demonstrate the *in vivo* relevance of these targets need to be strengthened.

Ced-3 mediated cleavage of LIN-28, LIN-14, and DISL-2 is an attractive hypothesis that is nicely supported by *in vitro* studies. What is most important is to substantiate this with *in vivo* studies, which the authors attempt using developmental westerns and transgene expression studies. As is, these experiments are weak. Figure 8 should ultimately have quantitation of LIN-28 signals vs actin, but the more problematic issue is that in Figure 4—figure supplement 1, the authors demonstrate that ced-3 animals have a developmental delay. Is it fair to compare animals at the same timepoint for Figure 8? E.g. at 24 hours wt are L3s but *ced-3* mutants are mostly L2s. Could the apparent persistence of LIN-28 in *ced-3* mutants be an indirect result? I.e. LIN-28 levels appear higher because the population is developmentally younger rather than lack of LIN-28 cleavage by CED-3?

To clarify the issue of the cleavage product not being detectable on westerns, is it possible to inactivate the N-end rule pathway to show the cleavage product is present in wt but not *ced-3* mutants?

Demonstration of physiological relevance is also important. The authors attempt this by engineering a mutation in the CED-3 cleavage site in a *lin-28::gfp* transgene and compare it to a similar strain with wild-type *lin-28::gfp* made years ago. They compare these strains using qPCR to indicate that copy number is similar. This is not terribly satisfying, as integrated arrays can be silenced to different degrees. At minimum, how does expression compare between these strains? A perhaps better experiment would be to integrate the constructs in single copy at the same locus or engineer the mutation into the endogenous *lin-28* locus.

The manuscript could also benefit from revision of the text. The presentation of data is sometimes hard to follow and suffers from some paragraphs that read as strings of facts without input of the authors' interpretations. There is also a tendency to throw in gene names without introduction or explanation of relevance. Both of these weaknesses make the paper less accessible to non-specialists. One example of an issue that warrants clarification is the description of phenotypes.

There are various places where phenotypes are referred to as “adult onset” or “more severe as animals progressed into adulthood”, but there isn't data that demonstrate these aspects of the phenotypes. A related issue is the use of “superficially wt adults”. For example, the rescue experiments (Figure 2) are difficult to interpret when the measurement is “superficially wt adults”. Does hypodermal or gut expression of *ain-1* rescue multiple phenes, restoring pleiotropic defects to superficially wild-type? Or does hyp expression only rescue hypodermal phenotypes? More explanation of the phenotype, perhaps in the methods, would help. Do the various aspects of the pleiotropic phenotype track together? Or do some animals have subsets? The multi-colored bar graphs in Figure 2 help some, but it is not completely clear how separable are the phenes.

[Editors’ note: further revisions were requested, as shown below.]

Thank you for resubmitting your work entitled “Non-apoptotic CED-3 activity regulates LIN-28 pluripotency pathway and others for robust development in *C. elegans*” for further consideration at *eLife*. Your revised article has been favorably evaluated by Detlef Weigel (Senior editor), a Reviewing editor, and the original two reviewers. The manuscript has been improved but there are some remaining issues that need to be addressed before acceptance, as outlined below:

In your revised manuscript entitled “Non-apoptotic CED-3 activity regulates LIN-28 pluripotency pathway and others for robust development in *C. elegans*,” you addressed most of the main concerns that the reviewers were concerned about. The assertions made in the current version of the manuscript are much more strongly supported than the previous version. The manuscript is more appropriately specific about phenotypes, having dispensed with “adult onset” and “superficially wild-type” in favor of more specific phenotypes. Additional data have been added to support your claim of LIN-28 as a proteolysis target of CED-3 and the conclusions have been toned down.

While the new experiments provide stronger evidence *ced-3* function in the regulation of miRNA targets, the text of the manuscript was difficult to read with regard to clarity and succinctness. In addition, the authors should also address the issues raised below in a corrected version:

1) Please provide an alternative title that would be understandable to a wider audience. ‘CED-3’ and ‘LIN-28’ may not be familiar to many people and some description might help.

2) The abstract should be clarified further as it implies that LIN-14, LIN-28, DISL-1 are all demonstrated in vivo targets.

3) It is appreciated that demonstration of CED-3-mediated proteolysis in vivo is challenging. The new data to support a role in proteolysis are helpful, but perhaps not definitive, a result that might be consistent with a mechanism that contributes to robustness of a developmental process. Analysis of *lin-28::gfp* in “age-matched” wt and *ced-3(lf)* mutants is provided through a picture of a head together with quantitation that indicates a modest increase in number of GFP+ cells when ced-3 is missing. The data seem to support the claim, but additional experimental details could help. What is the definition of “age-matched” here? Are the animals scored simply somewhere within the L3 stage? Or were they selected at a specific point within the stage by cell division pattern? This is an issue because the wild-type transgene is not always off; are the animals all at the same point in the L3? Also, are the cell types in question consistent with LIN-28 function?

4) The authors should avoid using developmental delay when they are referring to heterochronic phenotypes. This is especially important because the authors do measure both the growth rates and temporal patterning of development. It may be easier to just use temporal patterning when they reference heterochronic phenotypes.

5) Figure 6, it is relevant to know when the additional seam cells arise. The Methods section on scoring has been elaborated, but this information still appears to be lacking. Are the animals born with the correct number of seam cells and it gradually increases through extra L2-type proliferative divisions or is it late onset? This issue should be easy to clarify. Figure 6 should include the quantification of Wild Type seam cell numbers.

6) The paragraph outlining LIN-28, LIN-14 and DIS-1 regulation in *C. elegans* development should be more clearly written. This would include an outline of seam cell development, stage-specific expression of the genes and then the phenotypic consequences (with regard to seam cell number) associated with mutations in each of these genes. The impact of the synthetic interactions will be more impactful.

7) Regarding the text about the tetra-peptide sequence for CED-3 cleavage: these sites (numbers etc.) should be shown in each of the proteins. In addition, while true, the LIN-28 isoform lacks this putative site but is barely mentioned throughout the text. Its absence would have implications for future experiments and discussions.

8) Experiments outlined in Figure 9 should be more clearly described in the text or in the Figure legend. Is it the case that the percentage of animals reaching adulthood actually means the percentage of animals reaching adulthood after a defined time interval? What is that time interval?

The authors argue that aspects of these experiments support their model and some do, but some of the comparisons involve two variables (two different transgenes and +/- *ced-3*) coupled to potential developmental delays, which continue to complicate interpretations. Perhaps the new Figure 9 makes the best argument. Here, the animals are being assayed for adult cuticle, so must have reach a particular stage. And one can compare the effect of each transgene on the phenotype: if D31A starts out with lower expression than the wild-type transgene, one might expect it to have a weaker phenotype than the wild-type, yet it appears stronger, consistent with LIN-28 staying high longer. (It is a bit puzzling that the wt transgene does not cause gaps in alae, as this was indicated in Moss et al. '97.)

Since the LIN-28(D31A)::GFP transgene outlined in Figure 9 produces a ced-3(lf)-like phenotype (a slowing of animal growth), one would expect that the expression of LIN-28(D31A)::GFP would maintain inappropriate expression pattern to a similar amount/time (if not more than) the wild-type LIN-28::GFP reporter exhibits in ced-3(lf) animals. Why was this not done? It seems so obvious and so easy to do.

9) It seems incredibly surprising that the additional expression of LIN-28::GFP in ced-3 mutant backgrounds (increased expression in a very small number of cells; Figure 8) reflects the dramatic differences in the expression of the endogenous LIN-28 at a similar time point/developmental stage described in the experiments outlined in Figure 8. While the perdurance of LIN-28::GFP expression in *ced-3(lf)* mutants is supported by the in vivo evidence, these dramatic differences in scale of the phenotype should be addressed in the text.

---

## [Author Response]

*We are in principle interested in the work, because it potentially reveals a new and exciting mechanism for CED-3 action. Unfortunately, the claims made in the article and abstract, which imply a direct effect of the CED-3 protease on the stability of Lin-28 present only one possible explanation of many for the observed data. Demonstrating such a direct effect would, however, be essential for eventual acceptance of the work. We would minimally expect direct in vivo observations of Lin28 protein stability in the appropriate ced3 genetic background in properly staged animals*.

*The main set of experiments that are not clear revolve around the interpretation of the LIN-28 time course in wild-type and ced-3 mutants. There is an important facet to the interpretation of these experiments that may be explained by the trivial notion that ced-3 mutants develop more slowly than wild-type animals. The reviewers feel that this caveat could be addressed as you will see in the more detailed comments below*.

The editor/reviewers raised an important and complex issue.

We agree with the editor/reviewers that when you observe a change in development (slow progression, in this case) and a change in gene expression at the same time, it is difficult to determine which is the cause. This problem is actually quite common in analyzing gene expression during development, including in published literature.

However, given that previous work has established the dominant roles of *lin-28* and *lin-14* in regulating timing, and we have shown LIN-28, LIN-14 and DISL-2 are cleaved by CED-3 *in vitro*, it would be reasonable to make a suggestion out of the correlation data (Figure 8) that lack of the CED-3 activity on these developmental regulators contributed to the timing defect. Doing further *in vivo* analysis suggested by the reviewers has certainly help to strengthen this claim.

We added four sets of new experimental data to address this question:

As requested by the reviewers and editor, we quantified LIN-28::GFP expression in individual stage-matched worms for *ced-3(lf)* relative to *ced-3* wild-type animals by DIC microscopy (Figure 8). These data for *ced-3(lf)* suggest that the Western blot data shown in Figure 8 cannot mainly be explained by a younger cohort.

We have added new data showing that the CED-3-cleavage-resistant mutant form of LIN-28 [*i.e.* LIN-28(D31A)::GFP] in a *ced-3(wild-type)* background has a delayed down-regulation of LIN-28 relative to the wild-type LIN-28::GFP using Western blot (Figure 9). Because this CED-3-cleavage-resistant LIN-28 mutation also causes a slight delay in development of *ced-3(wild type)* worms (Figure 9), it provides a strong piece of evidence that lack of LIN-28 cleavage by CED-3 is one of the causes of the delayed developmental progression.

In this new Western blot (Figure 9), down-regulation of the wild-type LIN-28 transgene in *ced-3(lf)* worms seems to be delayed more than LIN-28(D31A) in wild type worms. Such a difference could be due to roles of CED-3 on other targets such as LIN-14 and DISL-2, which may cause a stronger developmental delay (Figure 9).

We added new data (Figure 9), as requested by reviewer 1, showing that the CED-3-cleavage-resistant mutant of LIN-28 [*i.e.* LIN-28(D31A)::GFP] in a *ced-3(wild-type)* background and the wild-type LIN-28::GFP reporter in a *ced-3(lf)* mutant background both have defective adult alae. These data support a clear function for CED-3-directed proteolysis of the LIN-28 protein in the heterochronic pathway.

Furthermore, we added new data (Figure 5), as requested by reviewer 1, showing that *ced-3(lf)* mutation enhances specific heterochronic phenotypes of *let-7-*family miRNA mutants and *ain-1(lf)* mutants, resulting in a significant number of animals with gapped adult alae.

Demonstrating *in vivo* protease activity of CED-3 on specific targets has always been difficult in the *ced-3* field. In our case, the challenge also lies on the fact that the CED-3-mediated repression of individual targets is subtle, as it works in concert with miRNAs and other proteolytic mechanisms, which is an important concept we try to deliver in this paper. We hope our new data and explanation here would be sufficient to address this issue.

In addition, we have modified the wording in the abstract and text to soften the conclusion regarding LIN-28 stability.

*It would also be important to provide a clear definition of what is being phenotypically assayed*.

We agree that clarity is important, as the editor/reviewers have suggested, and we have now revised all of the Figure panels in question to show all phenotypic details.

Reviewer #1:

*Summary: While the authors outline a nice genetic strategy to identify components that function with or in parallel to genes involved in general miRNA regulation and also clearly demonstrate that* ced-3 *functions in this capacity and outside its role in cell death, the experiments that demonstrate LIN-28 regulatory defects in* ced-3 *mutants are need expansion. The in vitro and genetic data are solid. Unfortunately, the phenotypes the statement that* ced-3 *contributes to the post-translational regulation of various miRNA targets is not complete. The authors have all of the reagents to prove this assertion follow-up experiments are outlined below.*

*Genetic screen and strategy: The authors present the findings of a genome-wide RNAi screen designed to identify factors that work together or in parallel to essential components of the miRNA machinery that regulates developmental gene expression. The basis for this screen relies on the fact that ain-1, encoding the C. elegans miRISC component/GW Protein homolog, is non-essential (due to a redundancy with the C. elegans ain-2 gene), displays a variety of pleiotropic phenotypes that can serve as the starting point for an enhancer screen. To identify genetic interactors, the authors systematically depleted each protein-coding gene individually by RNAi and scored for a wide range of general phenotypes. This resulted in the identification of 126 genes, that when depleted, result in a reproducibly synergistic enhancement of ain-1 and ain-2 mutant phenotypes. This list is composed of genes that function in a wide array of cellular and development functions and is interesting in itself with regard to the many functions for miRNAs in normal development*.

The suggestion to move the two panels from Figure 1 to Figure 2 is a good idea. These suggestions have been followed.

Reviewer #2:

*The manuscript by Weaver et al. presents the data from a carefully controlled genome-wide RNAi screen to search for genes whose depletion enhances phenotypes of animals compromised for miRISC function. This data set is likely to be of broad interest to the scientific community. A significant and somewhat surprising finding from this screen is the discovery that the CED-3 caspase has non-apoptotic roles in multiple developmental processes. Other recent studies have implicated* ced-3 *in similar such functions, but did not pursue* ced-3 *involvement to the level of protein targets. One key aspect of this paper is the identification of candidate* ced-3 *target proteins with bearing on the synthetic phenotype. However, the experiments to demonstrate the in vivo relevance of these targets need to be strengthened.*

Ced-3 *mediated cleavage of LIN-28, LIN-14, and DISL-2 is an attractive hypothesis that is nicely supported by in vitro studies. What is most important is to substantiate this with in vivo studies, which the authors attempt using developmental westerns and transgene expression studies. As is, these experiments are weak.*
Figure 8
*should ultimately have quantitation of LIN-28 signals vs actin, but the more problematic issue is that in*
Figure 4—figure supplement 1*, the authors demonstrate that ced-3 animals have a developmental delay. Is it fair to compare animals at the same timepoint for*
Figure 8*? E.g. at 24 hours wt are L3s but* ced-3 *mutants are mostly L2s. Could the apparent persistence of LIN-28 in* ced-3 *mutants be an indirect result? I.e. LIN-28 levels appear higher because the population is developmentally younger rather than lack of LIN-28 cleavage by CED-3?*

*To clarify the issue of the cleavage product not being detectable on westerns, is it possible to inactivate the N-end rule pathway to show the cleavage product is present in wt but not* ced-3 *mutants?*

The reviewer raises some important concepts. First, we agree and have now added quantitation of Figure 8 in the supplement. Second, regarding the “developmentally younger” concern, please see the accompanying response for new experimental data and explanations that address this point. Third, the reviewer brought up a very interesting question regarding whether we can test the interaction between *ced-3*, miRNA, and the N-end rule pathway. We have actually done several experiments addressing this issue and realized that it is not as simple in an animal system. To make a solid conclusion, we will need to treat this as a separate project showing both *in vivo* and *in vitro* data, as well as analyzing the functional interaction between the N-end rule pathway with *ced-3* and miRNAs. If successful, perhaps we could come back to add a short Research advance attached to this first paper. As for this paper, we have kept it as a hypothesis. We have modified the wording to soften the related statements.

*Demonstration of physiological relevance is also important. The authors attempt this by engineering a mutation in the CED-3 cleavage site in a* lin-28::gfp *transgene and compare it to a similar strain with wild-type* lin-28::gfp *made years ago. They compare these strains using qPCR to indicate that copy number is similar. This is not terribly satisfying, as integrated arrays can be silenced to different degrees. At minimum, how does expression compare between these strains? A perhaps better experiment would be to integrate the constructs in single copy at the same locus or engineer the mutation into the endogenous* lin-28 *locus.*

We appreciate this concern and have added quantified Western blot data from three independent replicates to show: 1) basal expression from the LIN-28(D31A)::GFP is somewhat less than the LIN-28::GFP alone which is consistent with the qPCR data measuring the transgene copy number (Figure 9—figure supplement 1), 2) the transgene with the point mutation alone had higher expression at 30 hours of feeding, suggesting that this mutation is sufficient to contribute to the developmental growth rate defect we observed.

Moreover, we agree that generating a single copy integration is ideal, which is now commonly practiced in our lab. Unfortunately, after many attempts, we were unable to generate such a line since the homologous rescue knock-in approach into either the endogenous locus or a Mos site requires high levels of the rescue construct that is unfortunately lethal. We are also aware of the two-step strategy of delivering constructs in sequential knock-in/replacement recombinations that could overcome lethality of the intact gene at high dosage. However, this would not only be time consuming but also low in feasibility because it requires that the first step creates a knock-out allele that would unfortunately generate the *lin-28(lf)* phenotype. This phenotype will render it unlikely for the second set of injections to generate transgenic F1s since *lin-28(lf)* animals would have a fully penetrant Pvl phenotype resulting in Egl with few progeny per adult. We have yet to reconcile this difficulty but feel that the transgenic lines we have used adequately test the concepts under consideration, including the new data assaying stage-matched animals, as well as observation of defects in adult-specific alae.

*The manuscript could also benefit from revision of the text. The presentation of data is sometimes hard to follow and suffers from some paragraphs that read as strings of facts without input of the authors' interpretations. There is also a tendency to throw in gene names without introduction or explanation of relevance. Both of these weaknesses make the paper less accessible to non-specialists. One example of an issue that warrants clarification is the description of phenotypes*.

We appreciate these points and have made modifications to improve the writing.

*There are various places where phenotypes are referred to as “adult onset” or “more severe as animals progressed into adulthood”, but there isn't data that demonstrate these aspects of the phenotypes. A related issue is the use of “superficially wt adults”. For example, the rescue experiments (*Figure 2*) are difficult to interpret when the measurement is “superficially wt adults”. Does hypodermal or gut expression of* ain-1 *rescue multiple phenes, restoring pleiotropic defects to superficially wild-type? Or does hyp expression only rescue hypodermal phenotypes? More explanation of the phenotype, perhaps in the methods, would help. Do the various aspects of the pleiotropic phenotype track together? Or do some animals have subsets? The multi-colored bar graphs in*
Figure 2
*help some, but it is not completely clear how separable are the phenes.*

We apologize for the cumbersome display of our results. To clarify our meaning, we have revised these statements from “adult-onset defects” to simply “defects” and clarified to “the frequency of abnormal phenotypes increased as the adults continued to age (Figure 2—figure supplement 2)…” As stated above, we were originally hesitant to show too much phenotypic detail as we felt this was distracting. Both reviewers and the editor seemed to have disliked this presentation of the data and we now include all phenotypic details. Due to the pleiotropic nature of the genetic defects, the specific phenotypes do not have a strict correlation with tissue-specific rescue.

[Editors’ note: further revisions were requested, as shown below.]

*1) Please provide an alternative title that would be understandable to a wider audience. ‘CED-3’ and ‘LIN-28’ may not be familiar to many people and some description might help*.

We have revised the title to: “CED-3 caspase acts with miRNAs to regulate non-apoptotic gene expression dynamics for robust development in *C. elegans*”.

*2) The abstract should be clarified further as it implies that LIN-14, LIN-28, DISL-1 are all demonstrated in vivo targets*.

We agree with this valid point and have made the changes in the abstract to more accurately summarize the data.

*3) It is appreciated that demonstration of CED-3-mediated proteolysis in vivo is challenging. The new data to support a role in proteolysis are helpful, but perhaps not definitive, a result that might be consistent with a mechanism that contributes to robustness of a developmental process. Analysis of* lin-28::gfp *in “age-matched” wt and* ced-3(lf) *mutants is provided through a picture of a head together with quantitation that indicates a modest increase in number of GFP+ cells when* ced-3 *is missing. The data seem to support the claim, but additional experimental details could help. What is the definition of “age-matched” here? Are the animals scored simply somewhere within the L3 stage? Or were they selected at a specific point within the stage by cell division pattern? This is an issue because the wild-type transgene is not always off; are the animals all at the same point in the L3? Also, are the cell types in question consistent with LIN-28 function?*

We understand the reviewers’ two main concerns presented here and we apologize for our lack of sufficient methodological detail. We addressed these questions by doing new experiments and improving the description.

First, we performed a new set of experiments to examine hypodermal expression at the L3 stage (new Figure 8). We first picked similarly sized L3 stage animals in a “blinded” fashion on a non-fluorescent dissecting scope. Prior to fluorescent illumination (to avoid bias), we measured gonad length to ensure they were very comparable (no significant difference found between the two strains: new Figure 8—figure supplement 2). It is already known that LIN-14, LIN-28 and miRNA pathway defects (with the exception of DAF-12) have only minor effects on gonad development making it a reliable marker to stage match animals (Ambros, 1997, *C. elegans* II; [38], Cell; and [1], Dev. Cell). Our method should thus provide a similar distribution of developmental sub-stages for both backgrounds within the L3 stage. We did not follow P cell division because we found that observation of P cell divisions was extremely difficult due to the roller phenotype of the animals. All animals were illuminated for 5 seconds for each picture by DIC optics. Multiple planes through the body were examined to ensure the brightest field of GFP positive cells was identified. Another person, who did not take the images, then quantified the GFP positive cells using Image J to obtain integrated values of GFP intensity. Data for all animals viewed by DIC were kept and reported.

Second, regarding the counting of GFP cells in the head for the data presented in the last revision (Figure 8), we did not sufficiently explain the method in the last version, which could have alleviated the concern. We selected animals in the blind and staged them as mentioned above. All animals were illuminated for 5 seconds for each picture by DIC optics. Multiple planes through the head were examined to ensure all GFP positive cells were identified. Another person, who did not take the images, then counted the number of GFP-positive cells. Data for all animals viewed by DIC were kept and reported. Those data were then presented in a way that we felt was most fair with respect to the null hypothesis (that “there was no difference between the two backgrounds”). However, analysis of the data showed that there was a significant difference in the number of cells still expressing GFP in the *ced-3(lf)* background. To address this concern, we have added the above details to the Methods section (Scoring LIN-28::GFP positive cells by DIC optics).

Third, we agree with the reviewer that lin-28 expression in the hypodermis (new data in Figure 8) is more relevant to the phenotypes of interest. However, we still decided to keep the neuron data but moved it to supplement (Figure 8—figure supplement 2). The function of *lin-28* in head neurons is currently unclear but the data still reflects the role of *ced-3* on *lin-28* expression.

*4) The authors should avoid using developmental delay when they are referring to heterochronic phenotypes. This is especially important because the authors do measure both the growth rates and temporal patterning of development. It may be easier to just use temporal patterning when they reference heterochronic phenotypes*.

We highly appreciate the suggestion including the helpful examples from the reviewer. We have modified the wording following the suggestions.

*5)*
Figure 6*, it is relevant to know when the additional seam cells arise. The Methods section on scoring has been elaborated, but this information still appears to be lacking. Are the animals born with the correct number of seam cells and it gradually increases through extra L2-type proliferative divisions or is it late onset? This issue should be easy to clarify.*
Figure 6
*should include the quantification of Wild Type seam cell numbers*.

This is a valid question and we have addressed it by doing additional experiments examining seam cells at three different stages. We find that the *ced-3(lf);ain-1(lf)* double mutant animals are born with the proper number of seam cells (new Figure 6—figure supplement 1). Analysis of third and fourth larval stages of the *ced-3(lf);ain-1(lf)* double mutant animals suggests that the supernumerary seam cells continue to develop during late larval stages (new Figure 6—figure supplement 1) though we cannot rule out earlier reiterations with an incompletely penetrant phenotype. We have also included the data for wild type animals (new Figure 6).

*6) The paragraph outlining LIN-28, LIN-14 and DIS-1 regulation in* C. elegans *development should be more clearly written. This would include an outline of seam cell development, stage-specific expression of the genes and then the phenotypic consequences (with regard to seam cell number) associated with mutations in each of these genes. The impact of the synthetic interactions will be more impactful.*

We have modified the writing following this helpful suggestion.

*7) Regarding the text about the tetra-peptide sequence for CED-3 cleavage: these sites (numbers etc.) should be shown in each of the proteins. In addition, while true, the LIN-28 isoform lacks this putative site but is barely mentioned throughout the text. Its absence would have implications for future experiments and discussions*.

We agree with these comments and have made the recommended changes. The putative cleavage sites for LIN-14 and DISL-2 are now added to the supplement. We have also added a discussion regarding the LIN-28B isoform.

*8) Experiments outlined in*
Figure 9
*should be more clearly described in the text or in the Figure legend. Is it the case that the percentage of animals reaching adulthood actually means the percentage of animals reaching adulthood after a defined time interval? What is that time interval*?

We apologize for our vague presentation of these data. We scored for the percentage of animals reaching adulthood by 96 hours after being added to OP50 food following synchronization in M9. We have now added these necessary details to the legend.

*The authors argue that aspects of these experiments support their model and some do, but some of the comparisons involve two variables (two different transgenes and +/- ced-3) coupled to potential developmental delays, which continue to complicate interpretations. Perhaps the new*
Figure 9
*makes the best argument. Here, the animals are being assayed for adult cuticle, so must have reach a particular stage. And one can compare the effect of each transgene on the phenotype: if D31A starts out with lower expression than the wild-type transgene, one might expect it to have a weaker phenotype than the wild-type, yet it appears stronger, consistent with LIN-28 staying high longer. (It is a bit puzzling that the wt transgene does not cause gaps in alae, as this was indicated in Moss et al. '97*.*)*

We understand the reviewers’ concern regarding the apparent inconsistency in not finding gapped adult alae in the wild-type LIN-28::GFP transgene. We feel the most likely explanation is simply an advance in camera technology. We only scored gaps that could be photographed by our camera (Zeiss Axiocam MRm). We have added the following statements to the Methods section: “Any thin region of alae that appeared as a gap through the oculars was imaged by the camera and evaluated on a large screen. Only alae observed as truly discontinuous by aid of the camera were scored as gapped. This method was applied equally to all strains throughout the study.” We have also added a note about the original phenotype (from [38]) to the Results section to help the reader. We would like to make note that the relative enhancement by *ced-3(lf)* or the D31A mutation alone is most significant.

*Since the LIN-28(D31A)::GFP transgene outlined in*
Figure 9
*produces a* ced-3(lf)*-like phenotype (a slowing of animal growth), one would expect that the expression of LIN-28(D31A)::GFP would maintain inappropriate expression pattern to a similar amount/time (if not more than) the wild-type LIN-28::GFP reporter exhibits in* ced-3(lf) *animals. Why was this not done? It seems so obvious and so easy to do.*

We appreciate the reviewers’ suggestion of examining the number of GFP positive cells in the D31A mutant by DIC optics, which we also initially set out to do. However, careful examination of the Western blot data shown in Figure 9 reveals that the basal expression level of the LIN-28(D31A)::GFP reporter is obviously lower than the wild-type transgene (quantified in Figure 9 0hr). Thus, a direct comparison of these lines by DIC microscopy would effectively be an uncontrolled experiment since there is no satisfactory way to control for exposure time between the two transgenic lines with differing basal expression. This is unlike the wild-type transgene compared in two genetic backgrounds: *ced-3(lf)* versus *ced-3(wt)* discussed above. Though not asked for here, we would like to state, as mentioned in the previous review rebuttal, that the toxicity of a lin-28-Cas9 rescue construct has prevented us from making single copy transgenes (which is not itself guaranteed to have the same level of expression). Moreover, we feel that examination of the adult-specific alae gaps is a more meaningful experiment (as suggested by the reviewers in the first review) since it is such a well-established phenotype linking persistent LIN-28 levels to defects in temporal cell fate patterning. This phenotype also considers the entire life history of the animal and thus any perdurance.

We added a paragraph to the Discussion to consider these concepts.

*9) It seems incredibly surprising that the additional expression of LIN-28::GFP in* ced-3 *mutant backgrounds (increased expression in a very small number of cells;*
Figure 8*) reflects the dramatic differences in the expression of the endogenous LIN-28 at a similar time point/developmental stage described in the experiments outlined in*
Figure 8*. While the perdurance of LIN-28::GFP expression in* ced-3(lf) *mutants is supported by the in vivo evidence, these dramatic differences in scale of the phenotype should be addressed in the text.*

We understand the reviewers’ concern about the difference in magnitude seen by the two methods. This difference may suggest that the observed fluorescence levels do not linearly reflect the protein levels. We have made note of it in the text. We would suggest that one should not expect that the dynamic range seen by Western blot is equal to that seen for direct fluorescence microscopy. Again, as noted by the reviewer, the *in vivo* defects in seam cell patterning and adult alae show that whatever the true alteration in expression level, it is sufficient to result in the observed defects. This has been emphasized in the Discussion section of the revised manuscript.